# Dynamically induced cascading failures in power grids

Benjamin Schäfer [1,2], Dirk Witthaut [3,4], Marc Timme[1,2] & Vito Latora [5,6]

Reliable functioning of infrastructure networks is essential for our modern society. Cascading failures are the cause of most large-scale network outages. Although cascading failures often exhibit dynamical transients, the modeling of cascades has so far mainly focused on the analysis of sequences of steady states. In this article, we focus on electrical transmission networks and introduce a framework that takes into account both the event-based nature of cascades and the essentials of the network dynamics. We find that transients of the order of seconds in the flows of a power grid play a crucial role in the emergence of collective behaviors. We finally propose a forecasting method to identify critical lines and components in advance or during operation. Overall, our work highlights the relevance of dynamically induced failures on the synchronization dynamics of national power grids of different European countries and provides methods to predict and model cascading failures.

[1] Chair for Network Dynamics, Center for Advancing Electronics Dresden (cfaed) and Institute for Theoretical Physics, Technical University of Dresden, 01062 Dresden, Germany. [2] Network Dynamics, Max Planck Institute for Dynamics and Self-Organization (MPIDS), 37077 Göttingen, Germany. [3] Forschungszentrum Jülich, Institute for Energy and Climate Research - Systems Analysis and Technology Evaluation (IEK-STE), 52428 Jülich, Germany. [4] Institute for Theoretical Physics, University of Cologne, 50937 Köln, Germany. [5] School of Mathematical Sciences, Queen Mary University of London, London E1 4NS, UK. [6] Dipartimento di Fisica ed Astronomia, Università di Catania and INFN, 95123 Catania, Italy. Correspondence and requests for materials should be addressed to B.Säf. (email: benjamin.schaefer@tu-dresden.de) or to M.T. (email: marc.timme@tu-dresden.de) or to V.L. (email: v.latora@qmul.ac.uk)

Our daily lives heavily depend on the functioning of many natural and man-made networks, ranging from neuronal and gene regulatory networks to communication systems, transportation networks and electrical power grids[1,2]. Understanding the robustness of these networks with respect to random failures and to targeted attacks is of outmost importance for preventing system outages with severe implications[3]. Recent examples, as the 2003 blackout in the Northeastern United States[4], the major European blackout in 2006[5] or the Indian blackout in 2012[6], have shown that initially local and small events can trigger large area outages of electric supply networks affecting millions of people, with severe economic and political consequences[7]. Cascading events will become more likely in the future due to increasing load[8] and additional fluctuations in the grid[9]. For this reason cascading failures have been studied intensively in statistical physics, and different network topologies and non-local effects have been considered and analyzed[10–18]. Complementary studies have employed simplified topologies that admit analytical insights, for instance in terms of percolation theory[19] or minimum coupling[20]. Results have shown, for instance, the robustness of scale-free networks[3,21,22], or the vulnerability of multiplex networks[23–26].

Although real-world cascades often include dynamical transients of grid frequency and flow with very well defined spatio-temporal structures, so far models of cascading failures have mainly focused on event-triggered sequences of steady states[13–16,27–30] or on purely dynamical descriptions of desynchronization without considering secondary failure of lines[31–35]. In particular, in supply networks such as electric power grids, which are considered as uniquely critical among all infrastructures[36], the failure of transmission lines during a blackout is determined not only by the network topology and by the static distribution of the electricity flow, but also by the collective transient dynamics of the entire system. Indeed, during the severe outages mentioned above, cascading failures in electric power grids happened on time scales of dozens of minutes overall, but often started due to the failure of a single element[37]. Conversely, sequences of individual line overloads took place on a much shorter time scale of seconds[4,5], the time scale of systemic instabilities, emphasizing the role of transient dynamics in the emergence of collective behaviors. For example, during the European blackout in 2006, a total of 33 high-voltage transmission lines tripped within a time period of 1 min and 20 s, with 30 of those lines failing within the first 19 s[5]. Notwithstanding the importance of these fast transients, the causes, triggers and propagation of cascades induced by transient dynamics have been considered only in a few works[10,38], and still need to be systematically studied[39]. Hence, we here focus on characterizing the dynamics of events that take place at the short time scale of seconds, which substantially contribute to the overall outages occurring in real grids.

This work complements the existing studies on cascading failures in power grids by linking nonlinear transient dynamics on short time scales to cascade events and simultaneously capturing line failures due to static overload. It is yet unrealistic to capture all aspects and time scales within a single model that is analytically tractable and provides mechanistic insights. Most of the previous studies[13–16,27–30,40], based on the analysis of sequences of steady states, consider the effects of power plant shutdown or line outages and did not take into account any transient dynamical effects at all. In contrast, a dynamical model might provide insights into cascading failures potentially induced on short time scales, thereby characterizing the time scales relevant to the majority of line failures.

In this article, we propose a general framework to analyze the impact of transient dynamics (of the order of seconds) on the outcome of cascading failures taking place over a complex network. Specifically, we go beyond purely topological or event-based investigations and present a dynamical model for electrical transmission networks that incorporates both the event-based nature of cascades and the properties of network dynamics, including transients, which, as we will show, can significantly increase the vulnerability of a network[10]. These transients describe the dynamical response of system variables, such as grid frequency and power flow, when one steady state is lost and the grid changes to a new steady state. Combining microscopic nonlinear dynamics techniques with a macroscopic statistical analysis of the system, we will first show that, even when a network seems to be robust because in the large majority of the cases the initial failure of its lines does only have local effects, there exist a few specific lines that can trigger large-scale cascades. We will then analyze the vulnerability of a network by looking at the dynamical properties of cascading failures. To identify the critical lines of the network we introduce and analytically derive a flow-based classifier that is shown to outperform measures solely relying on the network topology, local loads or network susceptibilities (line outage distribution factors). Finally, we find that the distance of a line failure from the initial trigger and the time of the line failure are highly correlated, especially when a measure of effective distance is adopted[41].

## Results

**The dynamics of cascading failures.** Failures are common in many interconnected systems, such as communication, transport and supply networks, which are fundamental ingredients of our modern societies. Usually, the failure of a single unit, or of a part of a network, is modeled by removing or deactivating a set of nodes or lines (or links) in the corresponding graph[42]. The most elementary damage to a network consists in the removal of a single line, since removing a node is equivalent to deactivate more than one line, namely all those lines incident in the node. For this reason, in the following of this work, we concentrate on line failures. In practice, the malfunctioning of a line in a transportation/communication network can either be due to an exogenous or to an endogenous event[43,44]. In the first case, the line breakdown is caused by something external to the network. Examples are the lightning strike of a transmission line of the electric power grid, or the sagging of a line in the heat of the summer. In the case of endogenous events, instead, a line can fail because of an overload due to an anomalous distributions of the flows over the network. Hence, the failure is an effect of the entire network.

Complex networks are also prone to cascading failures. In these events, the failure of a component triggers the successive failures of other parts of the network. In this way, an initial local shock produces a sequence of multiple failures in a domino mechanism which may finally affect a substantial part of the network. Cascading failures occur in transportation systems[45,46], in computer networks[47], in financial systems[48], but also in supply networks[23]. When, for some either exogenous or endogenous reason, a line of a supply network fails, its load has to be somehow redistributed to the neighboring lines. Although these lines are in general capable of handling their extra traffic, in a few unfortunate cases they will also go overload and will need to redistribute their increased load to their neighbors. This mechanism can lead to a cascade of failures, with a large number of transmission lines affected and malfunctioning at the same time. One particular critical supply network is the electrical power grid displaying for example large-scale cascading failures during the blackout on 14 August 2003, affecting millions of people in North America, and the European blackout that occurred on 4 November 2006. In order to model cascading failures in power transmission networks, we propose to use the framework of the

swing equation, see Eq. (14) in Methods, to evaluate, at each time, the actual power flow along the transmission lines of the network and we compare it to the actual available capacity of the lines, see Methods. Typical studies of network robustness and cascading failures in power grids adopted quasi-static perspectives[13–16,27–30] based on fixed-point estimates of the variables describing the node states. Such an approach, in the context of the swing equation, is equivalent to the evaluation of the voltage phase angles $\{\theta_i\}$ as the fixed-point solution of Eq. (14) or a power flow analysis[36]. In contrast, we use here the swing equation to dynamically update the angles $\theta_i(t)$ as functions of time, and to compute real-time estimates of the flow on each line. The flow on the line $(i, j)$, with coupling $K_{ij}$ at time $t$ is obtained as:

$$F_{ij}(t) = K_{ij}\sin\left(\theta_j(t) - \theta_i(t)\right). \tag{1}$$

Having the time evolution of the flow along the line $(i, j)$, we compare it to the capacity $C_{ij}$ of the line, i.e., to the maximum flow that the line can tolerate. There are multiple options how we can define the capacity of a line in the framework of the swing equation. One possibility is the following. The dynamical model of Eq. (14) itself would allow a maximum flow equal to $F_{ij} = K_{ij}$ on the line $(i, j)$. However, in realistic settings, ohmic losses would induce overheating of the lines which has to be avoided. Hence, we assume that the capacity $C_{ij}$ is set to be a tunable percentage of $K_{ij}$. In order to prevent damage and keep ground clearance[49,50], the line $(i, j)$ is then shutdown if the flow on it exceeds the value $\alpha K_{ij}$, where $\alpha \in [0, 1]$ is a control parameter of the model. The overload condition on the line $(i, j)$ at time $t$ finally reads:

$$\text{overload} : \left|F_{ij}(t)\right| > C_{ij} = \alpha K_{ij}. \tag{2}$$

Notice that the capacity $C_{ij} = \alpha K_{ij}$ is an absolute capacity, i.e., it is independent from the initial state of the system. This is different from the definition of a capacity relative to the initial flows, $\tilde{C}_{ij} := (1 + \alpha)F_{ij}(0)$, which has been commonly adopted in the literature[10,27,51].

Having defined the fixed point of the grid, given by the solution of Eq. (16), and the capacity of each line, we explore the robustness of the network with respect to line failures. We first consider the ideal scenario in which all elements of the grid are working properly, i.e., all generators are running as scheduled and all lines are operational. We say that the grid is $N - 0$ stable[52] if the network has a stable fixed point and the flows on all lines are within the bounds of the security limits, i.e., do not violate the overload condition Eq. (2), where the flows are calculated by inserting the fixed-point solution into Eq. (1).

Next, we assume the initial failure of a single transmission line. We call the new network in which the corresponding line has been removed the $N - 1$ grid. Since the affected transmission line can be any of the $|E|$ lines of the network, we have $|E|$ different $N - 1$ grids. If the $N - 1$ grid still has a fixed point for all possible $|E|$ different initial failures, and all of these fixed points result in flows within the capacity limits, the grid is said to be $N - 1$-stable[36,49,50]. While traditional cascade approaches usually test $N - 0$ or $N - 1$ stability using mainly static flows, our proposal is to investigate cascades by means of dynamically updated flows according to the power grid dynamics of Eq. (1). This allows for a more realistic modeling of real-time overloads and line failures. In practice, this means to solve the swing equation dynamically, update flows and compare to the capacity rule Eq. (2), removing lines whenever they exceed their capacity. Thereby, our $N - 1$ stability criterion demands not only the stable states to stay within the capacity limits but also includes the transient flows on all lines. See Supplementary Note 1 for details on our procedure, and Supplementary Note 6 for an investigation of the case of lines tripping after a finite overload time.

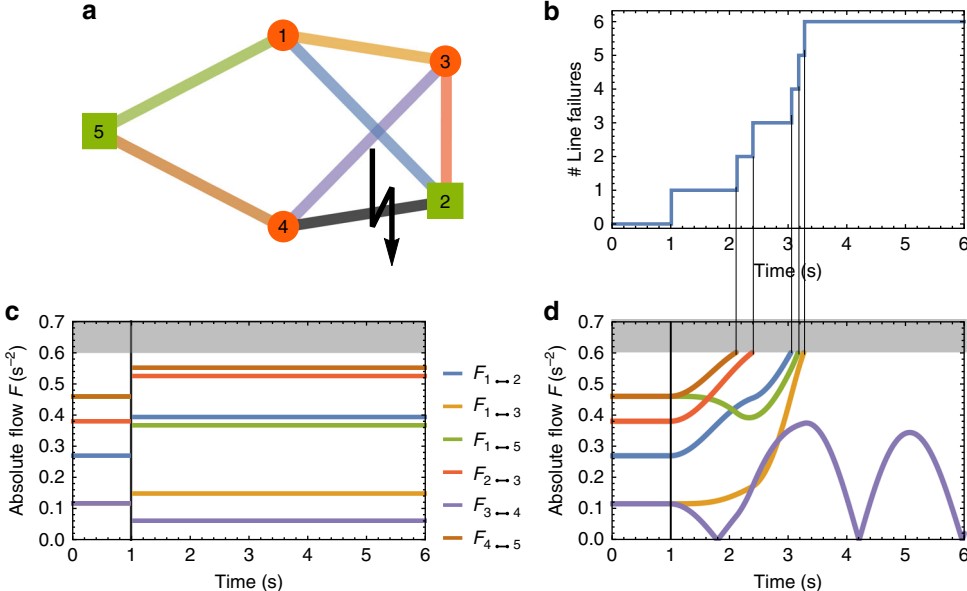

**Fig. 1** Dynamical overload reveals additional line failures compared to static flow analysis. **a** A five node power network with two generators $P^+ = 1.5\,\text{s}^{-2}$ (green squares), three consumers $P^- = -1\,\text{s}^{-2}$ (red circles), homogeneous coupling $K \approx 1.63\,\text{s}^{-2}$, and tolerance $\alpha = 0.6$ is analyzed. To trigger a cascade, we remove the line marked with a lightening bolt (2,4) at time $t = 1\,\text{s}$. Other lines are color-coded as the flows in **c** and **d**. **b** We observe a cascading failure with several additional line failures after the initial trigger due to the propagation of overloads. **c** The common quasi-static approach of analyzing fixed-point flows would have predicted no additional line failures, since the new fixed point is stable with all flows below the capacity threshold. **d** Conversely, the transient dynamics from the initial to the new fixed-point overloads additional lines which then fail as soon as their flows exceed their capacity (gray area)

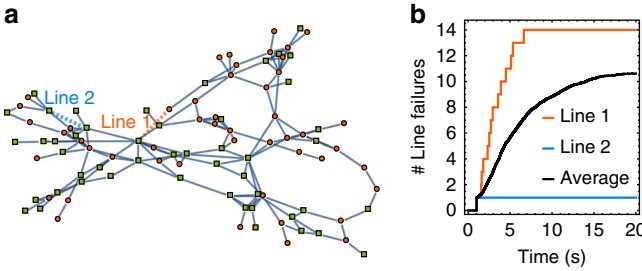

**Fig. 2** The effect of a cascade of failures strongly depends on the choice of the initially damaged line. **a** The network of the Spanish power grid with distributed generators with $P^+ = 1\,\text{s}^{-2}$ (green squares) and consumers with $P^- = -1\,\text{s}^{-2}$ (red circles), homogeneous coupling $K = 5\,\text{s}^{-2}$, and tolerance $\alpha = 0.52$ is analyzed. Two different trigger lines are selected. **b** The number of line failures as a function of time for the two different trigger lines highlighted in **a** and for an average over all possible initial damages. Some lines do only cause a single line failure, while others affect a substantial amount of the network. On average most line failures do take place within the first 20 seconds of the cascade

In order to illustrate how our dynamical model for cascading failures works in practice, we first consider the case of the network with $N = 5$ nodes and $|E| = 7$ lines shown in Fig. 1. We assume that the network has two generators, the two nodes reported as green squares, characterized by a positive power $P^+ = 1.5\,\text{s}^{-2}$, and three consumers, reported as red circles, with power $P^- = -1\,\text{s}^{-2}$. For simplicity we have adopted here a modified "per unit system" obtained by replacing real machine parameters with dimensionless multiples with respect to reference values. For instance, here a "per unit" mechanical power $P_{\text{per unit}} = 1\,\text{s}^{-2}$ corresponds to the real value $P_{\text{real}} = 100\,\text{MW}$[36,50]. Moreover, we assume homogeneous line parameters throughout the grid, namely, we fix the coupling for each couple of nodes $i$ and $j$ as $K_{ij} = Ka_{ij}$, with $K = 1.63\,\text{s}^{-2}$ and unweighted adjacency matrix $a_{ij}$. In order to prepare the system in its stable state, we solve Eq. (16) and calculate the corresponding flows at equilibrium. We then fix a threshold value of $\alpha = 0.6$. With such a value of the threshold, none of the flows is in the overload condition of Eq. (2), and the grid is $N - 0$ stable. Next, we perturb the stable steady state of the grid with an initial exogenous perturbation. Namely, we assume that line (2, 4) fails at time $t = 1$, due to an external disturbance. By using again the static approach of Eq. (16) to calculate the new steady state of the system, it is found that all flows have changed but they still are all below the limit of 0.6, as shown in Fig. 1c. Hence, with respect to a static analysis, the grid is $N - 1$-stable to the failure of line (2, 4). Despite this, the capacity criterion in Eq. (2) can be violated transiently, and secondary outages emerge dynamically. As Fig. 1d shows, this is indeed what happens in the example considered. Approximately one second after the initial failure, the line (4, 5) is overloaded, which causes a secondary failure, leading to additional overloads on other lines and their failure in a cascading process that eventually leads to the disconnection of the entire grid. The whole dynamics of the cascade of failures induced by the initial removal of line (2, 4) is reported in Fig. 1d. A dynamical update of the cascading algorithm is also shown in Supplementary Movie 1.

Dynamical cascades are not limited to small networks as the one considered in this example, but also appear in large networks. In order to show this, we have implemented our model for cascading failures on a network based on the real structure of the Spanish high-voltage transmission grid. The network is reported in Fig. 2 and has $N_{\text{Spanish}} = 98$ nodes and $|E|_{\text{Spanish}} = 175$ edges. We remark that, while we have complete knowledge of the

network topology, due to only partial information available on line parameters and power distribution, we have to estimate those missing parameters. Therefore, we have investigated several different power distributions and coupling scenarios in our simulations, including homogeneous versus heterogeneous coupling, as well as considering cases with many small power plants, compared to cases of fewer but larger plants. All parameter choices we have adopted are further specified in Supplementary Note 1 and the Data Availability Statement. We start by selecting a set of distributed generators (green squares), each with a positive power $P^+ = 1\,\text{s}^{-2}$, and consumers (red circles), with negative power $P^- = -1\,\text{s}^{-2}$. As in the case of the previous example, we adopt a homogeneous coupling, namely we fix $K_{ij} = Ka_{ij}$ with $K = 5\,\text{s}^{-2}$ for each couple of nodes $i$ and $j$. We also fix a tolerance value $\alpha = 0.52$, such that none of the flows is in the overload condition of Eq. (2), and the grid is initially $N - 0$ stable. We notice from the effects of cascading failures shown in Fig. 2 that the choice of the trigger line significantly influences the total number of lines damaged during a cascade. For instance, the initial damage of line 1 (dashed red line in Fig. 2a) causes a large cascade of failures with 14 lines damaged in the first seven seconds, while the initial damage of line 2 (dashed blue line in Fig. 2a) does not cause any further line failure, as the initial shock is in this case perfectly absorbed by the network. Figure 2 also displays the average number of failing lines as a function of time. Here, we average over all lines of the network considered as initially damaged lines. We notice that the cascading process is relatively fast, with all failures taking place within the first $T_{\text{Cascade}} = 20\,\text{s}$. This further supports the adoption of the swing equation, which is indeed mainly used to describe short time scales, while more complex and less tractable models are required to model longer times[50].

**Statistics of dynamical cascades**. To better characterize the potential effects of cascading failures in electric power grids, we have studied the statistical properties of cascades on the topology of real-world power transmission grids, such as those of Spain and France[53]. In particular, we have considered the two systems under different values of the tolerance parameter $\alpha$[27], and for various distributions of generators and consumers on the network. As in the examples of the previous section, we have also analyzed all possible initial damages triggering the cascade. To assess the consequences of a cascade, we have focused on the following two quantities. First, we analyze the number of lines that suffered an overload, and are thus shutdown during the cascading failure process. This number is a measure of the total damage suffered by the system in terms of loss of its connectivity. Second, we record the fraction of nodes that have experienced a desynchronization during the cascade, which represents a proxy for the number of consumers affected by a blackout, see Supplementary Note 1 for details on the implementation. In both the cases of affected lines and affected nodes, the numbers we look at are those obtained at the end of the cascading failure process.

Figure 3 shows the results obtained for the case of the network of the Spanish power transmission grid. The same homogeneous coupling and distribution of generators and consumers is adopted as in Fig. 2. We have considered each of the lines as a possible initial trigger of the cascade, and averaged the final number of line failures and unsynchronized nodes over all realizations of the dynamical process. We have repeated this for multiple values of the tolerance coefficient $\alpha$. As expected, a larger tolerance results in fewer line failures and fewer unsynchronized nodes, because it makes the overload condition of Eq. (2) more difficult to be satisfied. As we decrease the network tolerance $\alpha$, the total number of affected lines and unsynchronized nodes after the

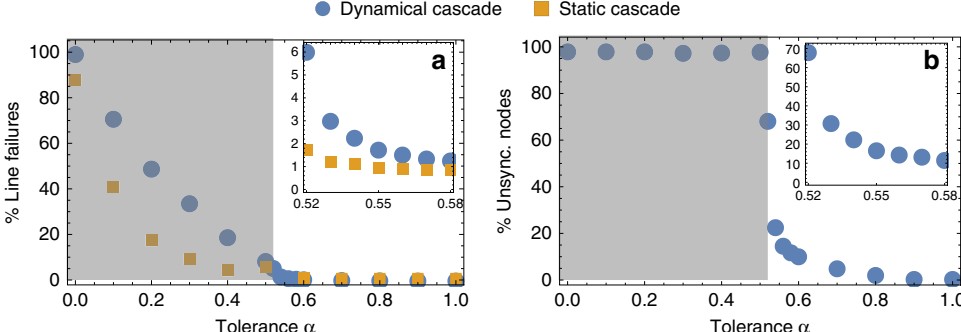

**Fig. 3** Effects of cascading failures in the Spanish power grid under different levels of tolerance. **a** The percentage of line failures in our model of cascading failures (circles), under different values of tolerance $\alpha$ is compared to the results of a static fixed-point flow analysis (squares). The static analysis largely underestimates the actual number of line failures in a dynamical approach. The difference between static and dynamical analysis is especially clear in the inset where we focus on the lowest values of $\alpha$ at which the network is $N - 0$ stable. The gray area is $N - 0$ unstable, i.e., the network without any external damage already has overloaded lines. **b** Percentage of unsynchronized (damaged) nodes after the cascade as a function of the tolerance $\alpha$. All analysis has been performed under the same distribution of generators and consumers as in Fig. 2, with homogeneous coupling of $K = 5\,\mathrm{s}^{-2}$

cascade suddenly increases at a value $\alpha \approx 0.5$, where we start to observe a propagation of the cascade induced by the initial external damage. Crucially, a dynamical approach, as the one considered in our model, identifies a significantly larger number of line failures (circles) compared to a static approach (squares). This is clearly visible in the inset of the left hand side of Fig. 3, where we zoom to the lowest values of $\alpha$ at which the network is $N - 0$ stable. For instance, at $\alpha = 0.52$ our model predicts that an average of six lines of the Spanish power grid are affected by the initial damage of a line of the network through a propagation of failures. Such a vulnerability of the network is completely unnoticeable by a static approach to cascading failures based on the analysis of fixed points. The static approach reveals in fact that on average only one other line of the network will be affected. We also note that the increase in the number of unsynchronized nodes for decreasing values of $\alpha$ is much sharper than that for overloaded lines. Below a value of $\alpha \approx 0.5$ the number of unsynchronized nodes jumps to 100%. This transition indicates a loss of the $N - 0$ stability of the system, meaning that, already in the unperturbed state several lines are overloaded according to the capacity criterion in Eq. (2) and thus fail. To study only genuine effects of cascades, in the following we restrict ourselves to the case $\alpha > 0.5$, where the grid is $N - 0$ stable, but not necessarily $N - 1$-stable. Furthermore, to assess the final impact of a cascade on a network, we mainly focus on total number of affected lines[4,5]. As discussed in the last section, damages to lines are indeed the most elementary type of network damages.

Furthermore, we have explored the role of centralized versus distributed power generation, and that of heterogeneous couplings $K_{ij}$, and also extended our analysis to other network topologies of European national power grids, namely those of France and of Great Britain, see Supplementary Note 2. In Fig. 4, we compare the results obtained for the Spanish network topology (three top panels) to those obtained for the French network (three bottom panels). With $N_{\mathrm{French}} = 146$ nodes and $|E|_{\mathrm{French}} = 223$ edges the French power grid is larger in size than the Spanish one considered in the previous figures ($N_{\mathrm{Spanish}} = 98$ and $|E|_{\mathrm{Spanish}} = 175$) and has a smaller clustering coefficient. In each case, we have calculated the total number of line failures at the end of the cascading failure when any possible line of the network is used as the initial trigger of the cascade. We then plot the probability of having a certain number of line failures in the process, so that the histogram reported indicates the size of the largest cascades and how often they occur. Notice that the probability axis uses a log-scale. For each network, we have

considered both distributed and centralized locations of power generators, and both homogeneous and heterogeneous network couplings. The centralized generation is thereby a good approximation to the classical power grid design with few large fossil and nuclear power plants powering the whole grid. In contrast, the distributed generation scheme describes well the case in which many small (wind, solar, biofuel, etc.) generators are distributed across the grid[31]. Finally, the choice of heterogeneous coupling is motivated by economic considerations, since maintaining a transmission network costs money and only those lines that actually carry flow are used in practice. In particular, we have worked under the following three different types of settings.

First, we consider distributed power and homogeneous coupling with an equal number of generators and consumers in the network, each of them having respectively $P^+ = 1\,\mathrm{s}^{-2}$ and $P^- = -1\,\mathrm{s}^{-2}$. The network uses homogeneous coupling with $K_{ij} = Ka_{ij}$ and $K = 5\,\mathrm{s}^{-2}$ for the Spanish (as in case of the previous figures) and $K = 8\,\mathrm{s}^{-2}$ for the French grid. Results for this case are shown in Fig. 4a and d. Next, we investigate centralized power and homogeneous couplings with consumers with $P^- = -1\,\mathrm{s}^{-2}$ and fewer but larger generators with $P^+ \approx 6\,\mathrm{s}^{-2}$. The network uses homogeneous coupling with $K = 10\,\mathrm{s}^{-2}$ for the Spanish and $K = 9\,\mathrm{s}^{-2}$ for the French grid. Results for this case are shown in Fig. 4b and e. Finally, we apply distributed power and heterogeneous coupling with homogeneous distribution of generators and consumers as in case 1. The network uses a heterogeneous distribution of the $K_{ij}$, so that the fixed-point flows on the lines are approximately $F \approx 0.5\,K$ both for the Spanish and the French grid, see Supplementary Note 1 for details. Results for this case are shown in Fig. 4c and f.

We chose each of the above settings such that no line is overloaded before the initial exogenous damage. We have performed simulations for two values of the tolerance parameter $\alpha$. For each of the two grids and of the three conditions above, the lowest value $\alpha = \alpha_1$ has been selected to be equal to the minimal tolerance such that each the network is $N - 0$ stable (yellow histograms). In addition, we have considered a second, larger value of the tolerance, $\alpha_2$, showing qualitatively different behaviors (blue histograms). As found in other studies[31–33,35], the (homogeneous) coupling $K$ has to be larger for centralized generation compared to distributed small generators to achieve comparable stability.

Initial line failures mostly do not cause any cascade and if they do, cascades typically affect only a small number of lines, see Fig. 4a. This means that the Spanish grid is in most of the cases $N$

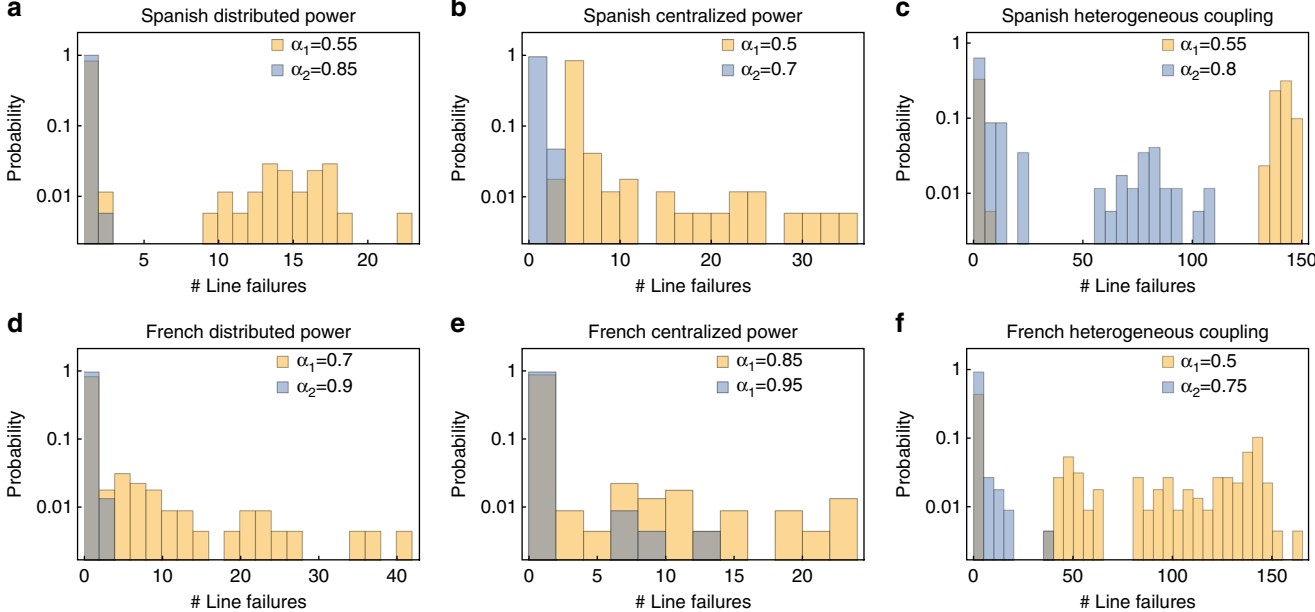

**Fig. 4** Network damage distributions in the Spanish and French power grids considering different parameter settings. The histograms shown have been obtained under three different settings. **a**, **d** The case of distributed power, i.e., equal number of generators and consumers, each with $P^+ = 1\,\mathrm{s}^{-2}$ and $P^- = -1\,\mathrm{s}^{-2}$, and homogeneous coupling with $K = 5\,\mathrm{s}^{-2}$ for the Spanish and $K = 8\,\mathrm{s}^{-2}$ for the French grid. **b**, **e** The case of centralized power, i.e., consumers with $P^- = -1\,\mathrm{s}^{-2}$ and fewer but larger generators with $P^+ \approx 6\,\mathrm{s}^{-2}$, and homogeneous coupling with $K = 10\,\mathrm{s}^{-2}$ for Spanish and $K = 9\,\mathrm{s}^{-2}$ for the French grid. **c**, **f** A case of distributed power as in **a** and **d**, but with heterogeneous coupling, so that the fixed-point flows on the lines are $F \approx 0.5\,K$ both for the Spanish and the French grid. For all plots we use two different tolerances $\alpha$, where the lower one is the smallest simulated value of $\alpha$ so that there are no initially overloaded lines ($N − 0$ stable)

$−1$-stable even in our dynamical model of cascades. Nevertheless, for $\alpha_1$, there exist a few lines that, when damaged, trigger a substantial part of the network to be disconnected. This leads to the question whether and how the distribution of generators or the topology of the network impact the size and frequency of the cascade. When comparing distributed (many small generators) in Fig. 4a to centralized power generation (few large generators) in Fig. 4b, we do not observe a significant difference in the statistics of the cascades. The same holds when comparing different network topologies, such as the Spanish and the French grid in Fig. 4d and e.

Conversely, allowing heterogeneous couplings introduces notable differences to emerge in Fig. 4c and f. To obtain heterogeneous couplings, we have scaled $K_{ij}$ at each line proportional to the flow at the stable operational state, see Supplementary Note 1. Thereby, we try to emulate cost-efficient grid planning which only includes lines when they are used. However, our results show that, under these conditions, the flow on a line with large coupling cannot easily be re-routed in our heterogeneous network when it fails[35]. For certain initially damaged lines, this leads to very large cascades in grids with heterogeneous coupling $K_{ij}$. For instance, both the Spanish and the French power grid show a peak of probability corresponding to cascades of about 150 line failures when $\alpha = \alpha_1$. But also in the case of $\alpha_2 = 0.8$, which corresponds to a $N − 1$-stable situation under the homogeneous coupling condition, the Spanish grid exhibits cascades involving from 50 to 100 lines in 5% of the cases under heterogeneous couplings, see Fig. 4c. The final number of unsynchronized nodes after the cascade, used as a measure of the network damage follows qualitatively a similar statistics. Namely, distributed and centralized power generation return similar statistical distributions of damage, while under heterogeneous couplings the system behaves differently. Furthermore, for each network, we have recorded the two extreme situations in which either all nodes or the grid stay synchronized, or the whole grid desynchronizes, see Supplementary Note 2.

What do the results obtained here imply about the robust operation of power grids? We have shown that a network that is initially stable ($N − 0$ stability), and remains stable even to the initial damage of a line ($N − 1$ stability) according to the standard static analysis of cascades, can display large-scale dynamical cascades when properly modeled. Although these dynamical overload events often have a very low probability, their occurrence cannot be neglected since they may collapse the entire power transmission network with catastrophic consequences. In the examples studied, we have found that some critical lines cause cascades resulting in a loss of up to 85% of the edges (Fig. 4c). Hence, it is extremely important to develop methods to identify such critical lines, which is the subject of the next section.

**Identifying critical lines**. The statistical analysis presented in the previous section revealed that the size of the cascades triggered by different line failures is very heterogeneous. Most lines of the networks investigated are not critical, i.e., they are either $N − 1$-stable even in our dynamical model of cascades, or cause only a very small number of secondary outages. However, for heavily loaded grids, as reported in Fig. 4, some highly critical lines emerge. Thereby, the initial failure of a single transmission line causes a global cascade with the desynchronization of the majority of nodes, leading to large blackouts. The key question here is whether it is possible to devise a fast method to identify the critical lines of a network. This might prove to be very useful when it comes to improving the robustness of the network.

In this section, we introduce a flow-based indicator for the onset of a cascade and demonstrate the effectiveness of its predictions by comparing them to results of the numerical simulation. In particular, we show that our indicator is capable of

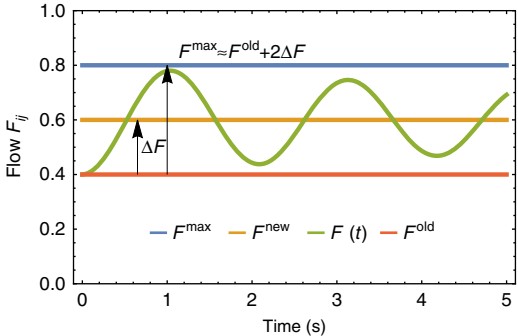

**Fig. 5** Introducing a flow-based estimator of the onset of a cascade. When cutting an initial line, the flows on a typical edge $(i, j)$ of the network increase from $F^{\mathrm{old}}$(red line) to $F^{\mathrm{new}}$ (orange line). On the basis of numerical observations, the transient flow $F(t)$ from the old to the new fixed point are well approximated as sinusoidal damped oscillation. Knowing the fixed-point flows, allows to compute the difference $\Delta F = F^{\mathrm{new}} - F^{\mathrm{old}}$ and estimate the maximum transient flow as $F^{\mathrm{max}} \approx F^{\mathrm{old}} + 2\Delta F$. This estimation is typically slightly larger than the real flow because the latter is damped

identifying the critical links of the network much better than other measures purely based on the topology or steady state of the network, such as the edge betweenness[11,27,28].

In order to define a flow-based predictor for the onset of a cascading failure, let us consider the typical time evolution of the flow along a line after the initial removal of the first damaged line $(a, b)$. As illustrated in Fig. 5, we observe flow oscillations after the initial line failure, which are well approximated by a damped sinusoidal function of time. See also Supplementary Note 4 for the time evolution of the flows for the case of the $N = 5$ node graph introduced in Fig. 1. Now, the steady flows of the network before and after the removal of the trigger line are obtained by solving Eq. (16) for the fixed-point angles $\{\theta_i^*\}$, which depend on the node powers $\{P_i\}$ and on the coupling matrix $\{K_{ij}\}$. Thereby, we obtain a set of nonlinear algebraic equations which have at least one solution if the coupling $K_{ij}$ is larger than the critical coupling[33]. For sufficiently large values of the coupling $K_{ij}$ there can be multiple fixed points[54]. In each case, we determine a single fixed point with small initial flows by using Newton's method, see Supplementary Note 1 for details. From the values of the fixed-point angles $\{\theta^*\}$ we calculate the equilibrium flow along each line, for instance line $(i, j)$, before and after the removal of the trigger line, from the expression:

$$F_{ij}^* = K_{ij} \sin\left(\theta_j^* - \theta_i^*\right). \qquad (3)$$

Let us indicate the initial flow along line $(i, j)$ in the intact network as $F_{ij}^{\mathrm{old}}$, and the new flow after the removal of the trigger line as $F_{ij}^{\mathrm{new}}$, assuming there still is a fixed point. Given enough time, the system settles in the new fixed point and the change of flow on the line is $\Delta F_{ij} = F_{ij}^{\mathrm{new}} - F_{ij}^{\mathrm{old}}$. Based on the oscillatory behavior observed in cascading events, see Fig. 5 for an illustration, we approximate the time-dependent flow on the line close to the new fixed point as:

$$F_{ij}(t) \approx F_{ij}^{\mathrm{new}} - \Delta F_{ij} \cos\left(\nu_{ij} t\right) e^{-Dt}, \qquad (4)$$

where $\nu_{ij}$ is the oscillation frequency specific to the link $(i, j)$ and $D$ is a damping factor. The maximum flow $F_{ij}^{\mathrm{max}}$ on the line

during the transient phase is then given by:

$$F_{ij}^{\mathrm{max}} \approx F_{ij}^{\mathrm{old}} + 2\Delta F_{ij}. \qquad (5)$$

Hence, for the cascade predictor we propose to test whether a line will be overloaded during the transient by computing $F_{ij}^{\mathrm{max}}$ from the expression above and by checking whether $F_{ij}^{\mathrm{max}}$ is larger than the available capacity $C_{ij}$ of the link. This procedure provides a good approximation of the real flows. However, it requires fixed-point calculations of the intact network and of the network after the initial trigger line is removed. Furthermore, it has to be repeated for each possible initial trigger line, so that a total of $|E| + 1$ fixed points is being computed, with $|E|$ being the number of edges. A possible way to simplify this procedure is to compute the fixed-point flows of the intact grid $F_{ij}^{\mathrm{old}}$ only, approximating the fixed-point flows after changes of the network topology by the Line Outage Distribution Factor (LODF)[17,18]. Details on this method can be found in Supplementary Note 1.

After starting the cascade by removing line $(a, b)$, we define our analytical prediction for the minimal transient tolerance $\left(\alpha_{ij}^{\mathrm{tr.}(a,b)}\right)_{\mathrm{min}}$ based on the maximum transient flow on line $(i, j)$ given in Eq. (5):

$$\left(\alpha_{ij}^{\mathrm{tr.}(a,b)}\right)_{\mathrm{min}} = \frac{F_{ij}^{\mathrm{max}}}{K_{ij}}, \qquad (6)$$

such that, if $\alpha > \left(\alpha_{ij}^{\mathrm{tr.}(a,b)}\right)_{\mathrm{min}}$, then cutting line $(a, b)$ as a trigger will not affect line $(i, j)$. Finally, we define the minimal tolerance $(\alpha^{\mathrm{tr.}(a,b)})_{\mathrm{min}}$ of the network as that value of $\alpha$ such that there is no secondary failure after the initial failure of the trigger line, i.e., the grid is $N - 1$ secure. We have:

$$\left(\alpha^{\mathrm{tr.}(a,b)}\right)_{\mathrm{min}} = \max_{(i,j)}\left(\alpha_{ij}^{\mathrm{tr.}(a,b)}\right)_{\mathrm{min}} = \max_{(i,j)}\left(\frac{F_{ij}^{\mathrm{max}}}{K_{ij}}\right), \qquad (7)$$

where the maximum is taken with respect to all links $(i, j)$ in the network and one trigger link $(a, b)$. If we set $\alpha \geq (\alpha^{\mathrm{tr.}(a,b)})_{\mathrm{min}}$ then, according to our prediction method, we expect no additional line failures further to the initial damaged line. Let us assume that the network topology is given, for instance that of a real national power grid, and that the tolerance level is preset due to external constrains like security regulations. Then, the calculation of $(\alpha^{\mathrm{tr.}(a,b)})_{\mathrm{min}}$ allows to engineer a resilient grid by trying out different realizations of $K_{ij}$. When changes of $K_{ij}$ are small, the new fixed-point flows are approximated by linear response of the old flows[17] giving us an easy way to design the power grid to fulfill safety requirements.

To measure the quality of our predictor for critical lines and to compare it to alternative predictors, we quantify its performance by evaluating how often it detects critical lines as critical (true positives) compared to how often it gives false alarms (false positives). In our model for cascading failures, a potential trigger line is classified as truly critical if its removal causes additional secondary failures in the network according to the numerical simulations of the dynamics[35]. The flow-based prediction is obtained by first calculating the minimal tolerance of the network $(\alpha^{\mathrm{tr.}(a,b)})_{\mathrm{min}}$ based on Eq. (7) and comparing it with the fixed tolerance $\alpha$ of a given simulation. If the obtained minimal tolerance is larger than the value of tolerance used in the numerical simulation, than the line is classified as critical by our predictor and additional overloads are to be expected. More

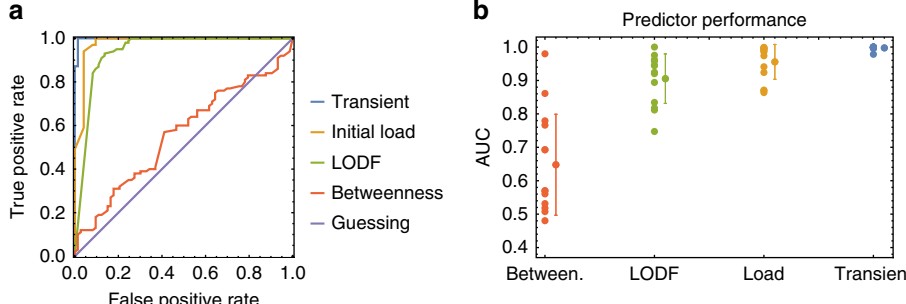

**Fig. 6** Comparing the predictions of the flow-based indicator of critical lines to other standard measures. Four different predictors are presented to determine whether a given line, if chosen as initially damaged, causes at least one additional line failures. Our dynamical predictor (indicated as Transient) is based on the estimated maximum transient flow (Eq. (5)). The predictor based on the LODF[17,18] uses the same idea but computes the new fixed flows based on a linearization of the flow computation. Predictors based on betweenness and initial load classify a line as critical if it is within the top $\sigma^{thr} \times 100\%$ of the edges with highest betweenness/load with threshold $\sigma^{thr} \in [0, 1]$. **a** The ROC curves obtained for the Spanish grid with heterogeneous coupling and tolerance $\alpha = 0.7$, while in **b** the AUC is displayed for all network settings presented in Fig. 4. For each predictor all individual scores are displayed on the left and the mean with error bars based on one standard deviation is shown on the right

formally, we use the following prediction rules:

$$\left( \alpha^{\mathrm{tr.}\,(a,b)} \right)_{\min} \geq \alpha + \sigma^{thr} \Rightarrow \text{critical}, \tag{8}$$

$$\left( \alpha^{\mathrm{tr.}\,(a,b)} \right)_{\min} < \alpha + \sigma^{thr} \Rightarrow \text{not critical}, \tag{9}$$

with a variable threshold $\sigma^{thr} \in [-1, 1]$, which allows to tune the sensitiviy of the predictor.

Analogously, we define a second predictor based on the LODF[17,18]. In this case, the expected minimal tolerance is obtained by approximating the new flow by the LODF, instead of computing them by solving for the new fixed points, see Supplementary Note 1.

We compare our predictors based on the flow dynamics to the pure topological (or steady-state based) measures that have been used in the classical analysis of cascades on networks. The idea behind such measures is the following. First, we consider the initial load on all potential trigger lines $(a, b)$: $L^{(a,b)} = |F_{ab}(t = 0)|$, i.e., the flow at time $t = 0$ on the line, when the system is in its steady state. Intuitively, highly loaded lines are expected to be more critical than less loaded ones. Hence, comparing each load $L^{(a,b)}$ to the maximum load on any line in the grid $L_{\max} := \max_{(i,j)} L^{(i,j)}$ leads to the following prediction:

$$L^{(a,b)} \geq \left( 1 - \sigma^{thr} \right) L_{\max} \Rightarrow \text{critical}, \tag{10}$$

$$L^{(a,b)} < \left( 1 - \sigma^{thr} \right) L_{\max} \Rightarrow \text{not critical}, \tag{11}$$

where $\sigma^{thr} \in [0, 1]$ is the prediction threshold.

Another quantity that is often used as a measure of the importance of a network edge is the edge betweenness[1,2]. The betweenness $b^{(a,b)}$ of edge $(a, b)$ is defined as the normalized number of shortest paths passing by the edge. A predictor based on the edge betweenness $b^{(a,b)}$ is then obtained by replacing $L^{(a,b)}$ by $b^{(a,b)}$ in the expressions above.

To evaluate the predictive power of the flow-based cascade predictors and to compare them to the standard topological predictors, we have computed the number of lines that cause a cascade by simulation and compared how often each predictor correctly predicted the cascade, thereby deriving the rate of correct cascade predictions (true positive rate) and rate of false alarms (false positive rate). These two quantities are displayed

in a receiver operator characteristics (ROC) curve, which reports the true positive rate versus the false positive rate when varying the threshold $\sigma^{thr}$. The ROC curve would go up straight from point (0, 0) to point (0, 1) in the ideal case in which the predictor is capable of detecting all real cascade events, while never giving a false positive. Conversely, random guessing corresponds to the bisector. Finally, any realistic predictor starts at the point (0, 0), i.e., never giving an alarm regardless of the setting, and evolves to the point (1, 1), i.e. always giving an alarm. The transition from (0, 0) to (1, 1) is tuned by decreasing the threshold $\sigma^{thr}$ determining when to give an alarm.

The ROC curves corresponding to the predictors introduced above are shown in Fig. 6a. A prediction based on the betweeness of the line is only as good as a random guess. In contrast, using the LODF and the initial load provide much better predictions. Finally, the analytical prediction outperforms any other method, well approximating an ideal predictor.

An alternative way to quantify the quality of a predictor is by evaluating the area under curve (AUC), that is the size of the area under the ROC curve. An ideal predictor would correspond to the maximum possible value AUC = 1, while a random guess produces an AUC of 0.5. So the closer the value of AUC for a given predictor is to 1, the better are the obtained predictions. AUC scores have been computed for different networks, settings and parameters. The results for the dynamical flow-based predictor, the predictor based on the LODF, as well as the initial load and betweenness predictors, are shown in Fig. 6b. The values of the AUC scores reported correspond to the different settings described in Fig. 4, allowing a more systematic comparison of predictors than that provided by a single ROC curve. Also from this figure it is clear that a prediction of the critical links based on their betweenness is on average only slightly better than random guessing. Furthermore, this result rises concerns on the indiscriminate use of the betweenness as a measure of centrality in complex networks. Especially when the dynamical processes of interest are well known, this must be taken into account in the definition of dynamical centrality measures for complex networks[11,55,56]. The LODF and initial load predictors perform relatively better on average, although they still display large standard deviations. This means that, for certain networks and settings they reach an AUC score close to the perfect value of 1, while in some other cases they only reach values of AUC equal to 0.8. Of these two indicators, the initial load predictor results are

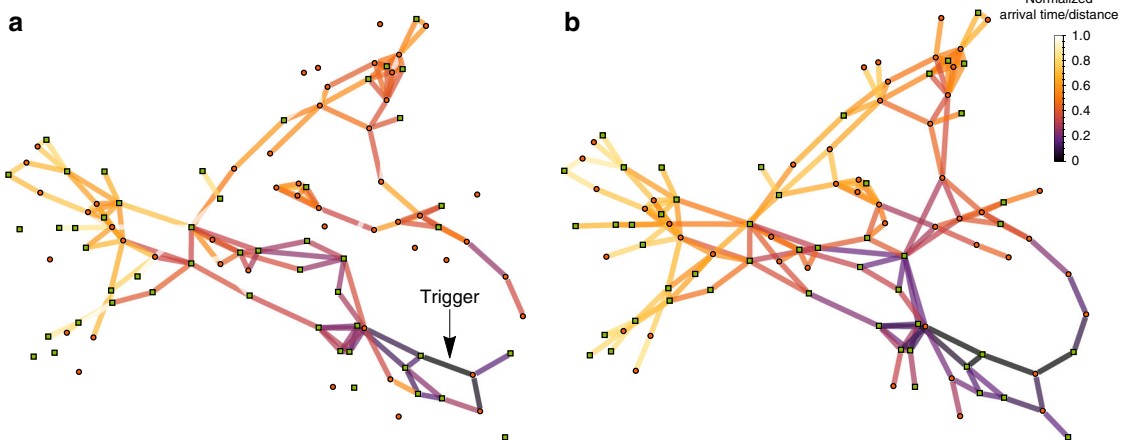

**Fig. 7** Mapping the propagation of a cascade on the Spanish power grid. **a** The edges of the network are color-coded based on the normalized arrival time of the cascade with respect to a specific initially damaged line, indicated as "Trigger" and **b** based on their normalized distance with respect to the trigger using the effective distance measure in Eq. (12). In both cases, darker colors indicate shorter distance / earlier arrival of the cascade. Normalization is carried out using the largest distance/arrival time. Edges that are not plotted are not reached by the cascade at all. The analysis has been performed using the Spanish grid with distributed generators with $P^+ = 1$ (green squares), consumers with $P^- = -1 \, s^{-2}$ (red circles), heterogeneous coupling and tolerance $\alpha = 0.55$

more reliable. Finally, our dynamical predictor, indicated in figure as "Transient" outperforms all alternative ones, in every single parameter and network realization. The figure indicates that the corresponding AUC scores reach values very close to 1. Moreover, this indicator displays the smallest standard deviation when different networks and parameter settings are considered. In conclusion, this seems to be the best indicator for the criticality of a link. However, the results show that, although the initial load predictor performs worse than our dynamical one, it might still be used when computational resources are scarce as it provides the second best predictions among those considered.

**Cascade propagation**. So far, we have shown that network cascades, i.e., secondary failures following an initial trigger, can well be caused by transient dynamical effects. We have proposed a model for power grids that takes this into account, and we have also developed a reliable method to predict whether additional lines can be affected by an initial damage, potentially triggering a cascade of failures. However, knowing whether a cascade develops or not does not answer another important question that is to understand how the cascade evolves throughout the network, and which nodes and links are affected and when. Intuitively, we expect that network components farther away from the initial failure should be affected later by the cascade. We have indeed observed that the time a line fails and its distance from the initial triggering link are correlated. Instead of merely using the graph topology to measure distances, we use a more sophisticated distance measure, the effective distance, based on the characteristic flow from one node to its neighbors. This idea has been first introduced in ref. [41] in the context of disease spreading, where the effective distance has been shown to be capable of capturing spreading phenomena better than the standard graph distance. The effective distance between two vertices $i$ and $j$ can be defined in our case as:

$$d_{ij} = 1 - \log\left(\frac{K_{ij}}{\sum_{k=1}^{N} K_{ik}}\right). \tag{12}$$

Here, we used the coupling matrix $K_{ij}$ as a measure of the flows between nodes[41]. All pairs of nodes not sharing an edge, i.e., such

that $K_{ij} = 0$, have infinite effective distance $d_{ij} = \infty$. At each node the cascade spreads to all neighbors but those that are coupled tightly, get affected the most and hence get assigned the smallest distance $d_{ij}$. Furthermore, the effective distance is an asymmetric measure, since $d_{ij} \neq d_{ji}$ in general. The quantity $d_{ij}$ is a property of two nodes, while the most elementary damage in our cascade model affects edges. Hence, the concept of distance has to be extended from couples of nodes to couples of links. For instance, in the case of an unweighted network, it is possible to define the (standard) distance between two edges as the number of hops along a shortest path connecting the two edges. In the case of a weighted graph, we make use of the measure of effective distance in Eq. (12) to define a distance between two edges as the minimal path length of all weighted shortest paths between two edges. The distance between two edges can then be obtained based on the definition of distances between nodes $\{d_{ij}\}$. Given the trigger edge $(a, b)$, the distance from edge $(a, b)$ to edge $(i, j)$ is given by:

$$d_{(a,b)\to(i,j)} = d_{ab} + \min_{v_1 \in \{a,b\}, v_2 \in \{i,j\}} d_{v_1 v_2}, \tag{13}$$

i.e., it is the minimum length of the shortest paths $a \to i$, $a \to j$, $b \to i$ and $b \to j$, plus the effective distance between the two vertices $a$ and $b$.

Figure 7 shows that the effective distance is capable of capturing well the properties of the spatial propagation of the cascade over the network from the location of the initial shock. The figure refers to the case of the Spanish grid topology with heterogeneous coupling (see Figs. 2 and 4). The temporal evolution of one particular cascade event, which is started by an initial exogenous damage of the edge marked as "Trigger", is reported. Network edges are color-coded based on the actual arrival time of the cascade in Fig. 7a, and compared to a color code based instead on their effective distance from the trigger line in Fig. 7b. Edges far away from the trigger line, in terms of effective distance, have brighter colors than edges close to the trigger. Similarly, lines at which the cascade arrives later are brighter than lines affected immediately. The figure clearly indicates that effective distance and arrival time are highly correlated, i.e., the cascade propagates throughout the network reaching earlier those edges that are closer according to the

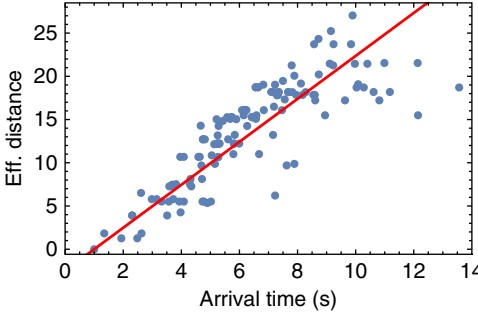

**Fig. 8** Effective distances between the initial trigger and secondary line outages are plotted as a function of arrival time. Each point in the plot represents one edge, while the straight line is the result of a linear fit. The reported fit indicates that the two quantities are related by an approximate linear relationship with regression coefficient $R^2 \approx 0.94$. Results refer to the Spanish power grid with the same parameters and trigger as used in Fig. 7

definition of effective distance. The relation between the effective distance of a line from the initial trigger and the time it takes for this line to be affected by the cascade is further investigated in the scatter plot of Fig. 8. We observe a substantial correlation between arrival time and shortest distance, indicating the possibility of an effective speed of cascading failures across the network. Averaging over all trigger lines, the correlation between cascade arrival time and effective distance, $\langle R^2_{\mathrm{eff.}} \rangle \approx 0.91$, is larger than between the arrival time and standard graph distance, $\langle R^2_{\mathrm{graph}} \rangle \approx 0.88$. See Supplementary Note 5 for details and ref. [57] for further discussion on propagation of cascades.

## Discussion

In this work, we have proposed and studied a model of electrical transmission networks highlighting the importance of transient dynamical behavior in the emergence and evolution of cascades of failures. The model takes into account the intrinsic dynamical nature of the system, in contrast to most other studies on supply networks, which are instead based on a static flow analysis. Differently from the existing works on cascading failures in power grids[10,13–16,27–30], we have exploited the dynamic nature of the swing equation to describe the temporal behavior of the system, and we have adopted an absolute flow threshold to model the propagation of a cascade and to identify the critical lines of a network. The differences with respects to the results of a static flow analysis are striking, as $N-1$ secure power grids, i.e., grids for which the static analysis does not predict any additional failures, can display large dynamical cascades. This result emphasizes the importance of taking dynamical transients of the order of seconds into account when analyzing cascades, and should be considered by grid operators when performing a power dispatch, or during grid extensions. Notably, our dynamical model for cascades not only reveals additional failures, but also allows to study the details of the spreading of the cascade over the network. We have investigated such a propagation by using an effective distance measure quantifying the distance of a line (link of the network) from the original failure, which strongly correlates with the time it takes for the cascade to reach this line. The observed correlation between propagation time and effective distance of a failure, points to the possibility of extracting an effective speed of the cascade propagation. This result may thus stimulate further research understanding propagation patterns on networks. Being able to measure the speed of a cascade would further contribute to the design of measures to stop or contain cascades in real-time because such

propagation speed determines how fast actions have to be taken. We remark that an approximately constant speed in terms of, e.g., the effective distance measure, may represent a highly non-local spreading in terms of geographical distances, also observed in[40,57]. Moreover, propagation patterns of line failures caused by current overloads, as investigated here, may be qualitatively different from those caused by voltage effects[4,58]. On longer time scales the operation of control systems and emergency measures such as load shedding must be taken into account to assess the impact of a cascade of failures. These features are typically studied in quasi-static models such that the short time scale considered in this paper offers a complementary view to the spreading of cascading failures.

While the swing equation is capable of capturing interesting dynamical effects previously unnoticed, it still constitutes a comparably simple model to describe power grids[50]. Alternative, more elaborated models would involve more variables, e.g., voltages at each node of the network to allow a description of longer time scales[59–62]. In addition, we only focused on the removal of individual lines in our framework, instead of including the shutdown of power plants, i.e., the removal of network nodes. These simplifications are mainly justified by the very same time scale of the dynamical phenomena. Most cascades observed in the simulation are very fast, terminating on a time scale of less than 10 s, which supports the choice of the swing equation[36,50]. Furthermore, such short time scales are consistent with empirical observations of real cascades in power grids, which were caused in a very short time by overloaded lines. Conversely, power plants (nodes of the network) were usually shutdown after the failure of a large fraction of the transmission grid. The same holds for load shedding, i.e., disconnecting consumers. Summing up, while the overall blackout takes place over minutes, critical damage is done within seconds due to line failures[4,5,7]. Hence, this article models the short time scale of line failures only.

In order to further support our conclusions, we have considered additional models and discussed the validity of the swing equation in Supplementary Note 3. In particular, we have also simulated a third order model that includes voltage dynamics, finding qualitatively similar results to those obtained with the swing equation. Furthermore, a recent study[63] also highlights that a DC approach misses important events, and an AC model is necessary to capture all aspects of cascades. While the authors in[63] use realistic (IEEE) grids and more detailed simulation models, we complement this numerical approach by providing semi-analytical insight into cascades. Specifically, we provide simple predictors of critical lines and observe a propagating cascade. Overall, our work indicates that a dynamical second order model, as the one adopted in our framework, is capable of capturing additional features compared to static flow analyses, while still making analytical approaches possible. This allows to go beyond the methods commonly adopted in the engineering literature, which are often solely based on heavy computer simulations of specific scenarios, e.g. ref. [64].

Furthermore, concerning the delicate issue of protecting the grid against random failures or targeted attacks, it is crucial to be able to identify critical lines whose removal might be causing large-scale outages. As we have seen, most of the lines of the networks studied in this article cause very small cascades when initially damaged. However, our results have also unveiled the existence in each of these networks of a few critical lines producing large outages, which in certain cases can even affect the entire grid. Within our modeling framework, we have been able to develop an analytical flow predictor that reliably identifies critical lines and outperforms existing topological measures in

terms of prediction power. As an alternative to the analytical flow predictor, when a faster assessment of criticality is required, the stable state flows of the intact grid can be used, although they are less reliable. We hope these two indicators can become a useful tool for grid operators to test their current power dispatch strategies against cascading threads.

In a time when our lives depend more than ever on the proper functioning of supply networks, we believe it is crucial to understand their vulnerabilities and design them to be as robust as possible. The results presented in this article represent only a first step in this direction and many interesting questions remain to be investigated and answered within our framework or similar approaches. If cascades often propagate non-locally, can a quantity like a propagation speed be defined or is it otherwise possible to predict cascade arrival times in a unified way? Which lines are affected by a large cascade, and which parts of the network are capable of returning to a stable state? What are the best mitigation strategies to contain a cascade or to stop its propagation? All these questions go beyond the scope of this article, whose main aim is to provide a first broad analysis of the importance of transients in the emergence and evolution of cascades. We hope the reported results will trigger the interest of the research community of physicists, mathematicians and engineers.

## Methods

**Modeling power grids**. When it comes to model the dynamics of a power transmission network, the swing equation is a simple way to deal with the key features of the system as a whole, namely its dynamical synchronization properties. Thereby, we avoid dealing directly with a complete dynamical description in terms of complicated power grid simulation software or static power-flow models which are routinely used to simulate specific scenarios on large-scale power grids by power engineers. The swing equation retains the dynamical features of AC power grids, by describing each of the elements of an electric power network as a rotating machine characterized by its angle and its angular velocity at a given time. In practice, a rotating machine either represents a large synchronous generator in a conventional power plant or a coherent subgroup, i.e., a group of strongly coupled small machines and loads which are tightly phase-locked in all cases. Note that this is a coarse-grained model where every node is modeled as a rotating machine with effective inertia. A node with higher demand than supply will then act as an effective consumer, i.e., a synchronous motor. The angle of each machine is assumed to be identical to the angle of the complex voltage vector, so that the angle difference of two machines determines the power flow between them to transport, for example, energy from a generator to a consumer.

More formally, let us suppose to have $N$ rotatory machines, each corresponding to a node of a network. Each machine $i$, with $i = 1, 2, …, N$ is characterized by its mechanical rotor angle $\theta_i(t)$ and by its angular velocity $\omega_i := d\theta_i/dt$ relative to the reference frame of $\Omega = 2\pi(50$ or $60)$ Hz. Furthermore, machine $i$ either feeds power into the network, acting as an effective generator with power $P_i > 0$, or absorbs power, acting as an effective consumer (corresponding to the aggregate consumers of an urban area) with power $P_i < 0$. The swing equation reads[34,50,65]:

$$\frac{d}{dt}\theta_i = \omega_i, \tag{14}$$

$$I_i \frac{d}{dt}\omega_i = P_i - \gamma_i \omega_i + \sum_{j=1}^{N} K_{ij} \sin\left(\theta_j - \theta_i\right), \tag{15}$$

where $\gamma_i$ is the damping of an oscillator, $I_i$ is the inertia constant and $K_{ij}$ is a coupling matrix governing the topology of the power grid network, and the strength of the interactions. In the following, we will both consider heterogeneous coupling $K_{ij}$ or we will assume homogeneous coupling $K_{ij} = K a_{ij}$, where $a_{ij}$ are the entries of the unweighted adjacency matrix that describes the connectivity of the network. For simplicity, we assume homogeneous damping $\gamma_i = \gamma$ and inertia $I_i = 1$ for all $i \in 1, …, N$. To derive Eq. (14) one has to assume that the voltage amplitude $V_i$ at each nodes is time-independent, that ohmic losses are negligible and that the changes in the angular velocity are small compared to the reference $\omega_i \ll \Omega$, see e.g, refs. [36,65], for details. All these assumptions are fulfilled as long as we model short time scales on the high-voltage transmission grid[50] which will be sufficient for our study. The coupling matrix $K_{ij}$ is an abbreviation for $K_{ij} = B_{ij} V_i V_j$ where $B_{ij}$ is the susceptance between two nodes[36]. The swing equation is especially well suited to describe the power grid dynamics on short time scales, as they appear in typical large-scale power grid cascades[4,5,7], however, we also

discuss other models returning qualitatively similar results in Supplementary Note 3.

The desired stable state of operation of the power grid network is characterized by all machines running in synchrony at the reference angular velocity $\Omega$, i.e., $\omega_i = 0 \; \forall i \in \{1, …, N\}$, implying $\sum_i P_i = 0$. Thereby, we determine the fixed point by solving for the angles $\theta_i^*$ in:

$$0 = P_i + \sum_{j=1}^{N} K_{ij} \sin\left(\theta_j^* - \theta_i^*\right). \tag{16}$$

The grid in its synchronous state is phase-locked, i.e., all angle differences do not change in time. This is important since the angle difference determines the flow along a line, and fluctuating angle differences would imply fluctuating conducted power which can in turn lead to the shutdown of a plant[36,50]. Furthermore, transmission system operators demand the frequency to stay within strict boundaries to ensure stability and constant phase locking[66].

Phase-locking and other synchronization phenomena arise in many different domains and applications, and have attracted the interest of physicists across fields[67]. One of the simplest synchronization models is the Kuramoto model which has been used, among other applications, to describe synchronization phenomena in fireflies, chemical reactions and simple neuronal models[68–70]. The swing equation shows similarities with the Kuromoto model, including the sinusoidal form of the coupling function and the existence of a minimal coupling threshold to achieve synchronization[33]. However, the swing equation includes a second derivative due to the inertial forces in the grid. Both equations share the same fixed points but the swing equations display dissipative forces and limit cycles that are not present in the Kuramoto model.

**Data availability**. The networks used in this study and examples of elementary cascades are provided at https://osf.io/jz4m6/. All data that support the results presented in the figures of this study are available from the authors upon request.

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

## Acknowledgements

We thank Vittorio Rosato for providing the national grid topologies. B.S. and V. L. acknowledge support from the EPSRC project EP/N013492/1, "Nash equilibria for load balancing in networked power systems". We gratefully acknowledge support from the Federal Ministry of Education and Research (BMBF Grant No. 03SF0472A-F), the Helmholtz Association (via the joint initiative "Energy System 2050—A Contribution of the Research Field Energy" and the Grant No.VH-NG-1025 to D.W.), the Göttingen Graduate School for Neurosciences and Molecular Biosciences (DFG Grant GSC 226/2 to B.S.) and the Max Planck Society (to M.T.). Finally, we gratefully acknowledge support from the German Science Foundation (DFG) by a grant toward the Cluster of Excellence "Center for Advancing Electronics Dresden" (cfaed)

## Author contributions

B.S. and V.L. designed the research. B.S. performed most simulations and generated figures. B.S, V.L, D.W., and M.T. contributed to discussing intermediate and final results and writing the paper.

## Additional information

**Competing interests:** The authors declare no competing interests.

