## [Peer Review File · Nature Communications]

Reviewers' comments:

Reviewer #1 (Remarks to the Author):

The manuscript describes a study of cascading failures in power grids in which a transmission line failure is modeled by the removal of the line and the voltage phase dynamics between failures is modeled by the swing equation. One conclusion drawn is that the swing-equation-based model may predict a cascading failure even when a static model based on the power flow equation does not. Two computationally efficient methods for identifying the initial triggers that cause cascades is proposed with their accuracy computationally validated. The time it takes for a cascade to propagate and reach a given line is shown to correlate with the effective distance between the given line and the initial triggering line.

I certainly do not doubt the importance of dynamical transient in understanding and predicting power grid cascades. However, I have identified four major issues that prevent me from recommending the manuscript for publication:

1) The swing equation [Eq. (1), which is based on the so-called classical model] is supposed to be valid only for periods on the order of one second [see Anderson and Fouad, Power system control and stability (IEEE Press, 2nd ed., 2003) for example]. Thus, drawing conclusions based on 50-sec simulations on cascades that can evolve over periods up to 10 sec is not well justified. Even when the 3rd-order model that includes the voltage magnitude dynamics is used, it is unclear whether this short-time-scale limitation can be lifted. Do Refs. [4, 20-23], cited for the 3rd-order model in Supp. Note 3 (I could not find any mention of the model in Ref. [4], though), justify the use of the model to accurately simulate the transient dynamics for periods of length at least 10 sec?

2) Large-scale cascades involve failures that occur on time scales much larger than the simulation time (50 sec) considered in this manuscript, as is described in, e.g., Ref. [4] on the 2003 US blackout. It may be that many of such cascades involve a portion in which fast dynamics induces multiple line failures, but if the goal of the study is to address just that portion, it needs to be clearly stated upfront and the interpretation/conclusions should be adjusted to reflect the scope of the study. In particular, the ultimate size of the damage can only be determined by including the power plant shutdown and load shedding (which occurs on much longer time scales), so what is analyzed throughout the manuscript is an unknown fraction of the damage caused by cascades.

3) The choice of parameters, in particular, P_j and K_{ij} , seems too far from reality. Does the homogeneous choice represent any realistic situation for the Spanish grid, for example? Having a

sizable generation or load on every node does not sound realistic, given that there are substations that just redistribute power coming in from one transmission line to others, and thus do not produce or consume power (if all such substations have been eliminated to generate the model, it should be stated, explained, and justified in the manuscript). These, together with all the other simplifying assumptions leading to Eq. (1), make the model used in this study too unrealistic to support the conclusions drawn and implications stated for real power grids.

4) I do not believe it is accurate to say that cascades studied here propagate "at a nearly constant propagation speed" with respect to the effective distance. The results in the manuscript show that the effective distance between the given line and the initial triggering line is correlated with the propagation time. However, this is not a sufficiently strong support for the description of the cascade dynamics as nearly constant propagation along the network. I believe a cascade often jumps a relatively long distance in a single step due to non-local rerouting of power flows after a line removal, and this substantially deviates from the image of a cascade wave front propagating at a constant speed (even with significant fluctuations around the mean speed).

Reviewer #2 (Remarks to the Author):

I think the paper provides a significant advance in modeling and understanding cascading failures in the grid by employing models based on "actual" grid dynamics.

Recent literature in complex systems/statistical physics/networks is saturated with works claiming to model the grid and proposing to mitigate cascading failures, building on "models" which have absolutely nothing to do with the underlying dynamics of the grid (that is, conserved flows: AC model or DC approximation at its basic level).

The present paper considers the AC model (swing equations), which, first and foremost, governs the actual dynamics of flows and phases on the grid. Further, this system of equations also shares some similarities with the Kuramoto model for synchronization, and thus the findings contribute to the broader understanding of synchronization dynamics.

Among the main findings is that static models, in general, can significantly underestimate the size of the cascades, thus understanding the full dynamic behavior of the grid (transients on short-time scales) is important to predict and mitigate cascading failures.

Also, the authors propose a flow-based predictor to identify critical lines, and an effective distance to track the propagation of the cascading failures, which seem to be working reasonably well to characterize lines and the propagation of cascades.

One comment I have is that I'm aware of some recent (independent) work of a group which have been working similar topics, namely the comparison of cascading failures in AC and DC power flow models in the grid:

Hale Cetinay, Saleh Soltan, Fernando A. Kuipers, Gil Zussman, and Piet Van Mieghem,

"Comparing the Effects of Failures in Power Grids under the AC and DC Power Flow Models",

(Under Review at IEEE Transactions on Network Science and Engineering)

<http://wimnet.ee.columbia.edu/wp-content/uploads/2017/05/comparing-the-effects-of-failures-in-power-grids-under-the-AC-and-DC-power-flow-models.pdf>

Perhaps would be best if the authors reach out to the authors above for publication information of the paper above, and mutually cite each others' work before final publication.

My additional minor comments are:

page 7:

when the authors discuss the various scenarios 1., 2., 3. (for distributed or centralized power distribution with homogeneous or heterogeneous couplings), and the discussion that follows in the right column, they should indicate what Figure the panels (a),(b),(c),(d) refer to. (Figure 4, I assume.)

title:

I think the model and the problem the authors study here is quite specific to the power grid (AC model/swing equations).

The title with the term "supply networks" sounds a bit overly general. Perhaps replacing it with "power grids" would be more in synch with the actual system studied.

In summary, I recommend the paper for publication with the above small changes/comments.

Reviewer #3 (Remarks to the Author):

The authors present an extensive analysis of cascading failures in power grids via the simplified swing-equation model, thus taking into account also the dynamics. The networks analyzed are high voltage power networks, i.e. transmission networks. The paper's approach is in the spirit of statistical mechanics, hence does not pretend to give a rigorous, final answer to the problem of black-outs in real electric system but tries to understand what can trigger a systemic failure. For the network configurations and parameters analyzed, the paper shows that simulations based on event-triggered sequences of steady states tend to predict a lower number of failures respect to the more extended dynamic model of the paper. Overall, the paper is very interesting and worth of publications.

I have just few of remarks for the authors

A) the term "supply networks" is misleading and recalls too much "supply chains" that themselves have been modeled via complex networks. Speaking of electric power grid, the correct terms are TRANSMISSION (high voltage, national & supra-national networks) and DISTRIBUTION (medium voltage, regional or big cities). The networks analysed in the paper are thus TRANSMISSION NETWORKS.

B) The term transient can be also misleading if the paper is to be read by a more technical audience (power engineers and system operators). In fact, they would tend to associate a transient to something that happens at a breaker in 20-30 cycles (half a second), i.e. a factor 10 respect to the analysis of the paper; even worse, the dynamics of a transient in 20 cycles is sharp and contains higher frequencies, hence by eye inspection the time factor would be > 100 .

D) A breaker does not trip a line instantaneously, it normally can withstand currents of order 1% of the rated currents for periods up to 1-3s. Hence, tripping a line when power hits the capacity is ok unless the derivative is flat. This point should not change the results, but perhaps could be cited.

E) paper [50] by Pahwa et al accounts for a cascading model amenable also of a mean field approximation; such mean field model has also been extended to interacting cascading systems in:

Cascades in interdependent flow networks

A Scala, PGDS Lucentini, G Caldarelli, G D'Agostino

Physica D: Nonlinear Phenomena 323, 35-39

that should be added to the references in "vulnerability of multiplex networks [18-20]"

Moreover, ref.

"non-local effects have been considered and analyzed [8-13]"

should be changed in

"non-local effects have been considered and analyzed [8-13,50]"

and

"event-triggered sequences of steady states [8, 10, 11, 21-24]"

should be changed in

"event-triggered sequences of steady states [8, 10, 11, 21-24,50]"

REPLIES TO THE COMMENTS OF THE REVIEWERS

REVIEWER #1:

Reviewer #1 *The manuscript describes a study of cascading failures in power grids in which a transmission line failure is modeled by the removal of the line and the voltage phase dynamics between failures is modeled by the swing equation. One conclusion drawn is that the swing-equation-based model may predict a cascading failure even when a static model based on the power flow equation does not. Two computationally efficient methods for identifying the initial triggers that cause cascades is proposed with their accuracy computationally validated. The time it takes for a cascade to propagate and reach a given line is shown to correlate with the effective distance between the given line and the initial triggering line. I certainly do not doubt the importance of dynamical transient in understanding and predicting power grid cascades. However, I have identified four major issues that prevent me from recommending the manuscript for publication:*

Our response: We thank the reviewer for the careful review of our paper. He/she positively summarizes our results, and in particular emphasizes our novel identification of critical links and propagation of cascades. However, the reviewer raises four points that prevented him/her from recommending the manuscript for publication. In the following we answer each of these four points in detail.

Reviewer #1: Point 1 *The swing equation [Eq. (1), which is based on the so-called classical model] is supposed to be valid only for periods on the order of one second [see Anderson and Fouad, Power system control and stability (IEEE Press, 2nd ed., 2003) for example]. Thus, drawing conclusions based on 50-sec simulations on cascades that can evolve over periods up to 10 sec is not well justified. Even when the 3rd-order model that includes the voltage magnitude dynamics is used, it is unclear whether this short-time-scale limitation can be lifted. Do Refs. [4, 20-23], cited for the 3rd-order model in Supp. Note 3 (I could not find any mention of the model in Ref. [4], though), justify the use of the model to accurately simulate the transient dynamics for periods of length at least 10 sec?*

Our response: The reviewer questions the use of the swing equation for cascades that can evolve over periods up to 10 seconds, as the swing equation is supposed to be valid for time scales of the order of seconds only. First, we note that most overloads, including the ones we predict, occur within the first few seconds after the initial failure. See for example Fig. 1, where the last overload occurs at time equal to 3 seconds, i.e., only 2 seconds after the initial failure. Even more, looking at the 2006 Europe outage [1] there were at least 14 outages in 14 seconds. Looking at the 2003 US outage there were six outages within 2 seconds [2].

One possibility to overcome short time scales limitations is to go to higher order models. For such reasons, we investigated a 3rd order model in the SI, showing that it gives qualitatively similar results to those obtained with the swing equation. In addition to this, Auer. et al [3] (namely, Ref. [20] in the previous version of our SI) have shown that models, such as the 4th order model, give very similar transient behaviors as the swing equation, and only differ in their long-term asymptotic dynamics. Since our analysis is mostly concerned with the transient dynamics on the scale of seconds, we think that higher order effects, as also discussed by Machowski et al. in Chapter “Advanced Power System Modelling”, Section “Synchronous Generator Models” of Ref. [4] (Ref. [4] in the previous version of our SI) would not alter our results.

Another modeling possibility is to investigate standard test grids, like IEEE test grids, by using fully realistic line parameters, but this can only be done numerically, as for example in the work by Cetinay et al. [5] (see also point 2 of Reviewer #2). However, in our paper we were interested in obtaining (semi-)analytical insight into cascades. Using models of very high detail, makes such analytical approaches much harder, if not impossible. Hence, we had to work with the swing equation (classical model), which still allows analytical treatment, compared to higher order models, and we believe it still captures the most important dynamical features of the grid for the first few seconds.

To make our modeling assumptions more explicit to the reader we have added a discussion of the validity of the swing equation, and of alternative and more detailed models in the revised version of the main text (blue highlighted text in the “Discussion” section), and in SI (in a brand new part of Supplementary Note 3).

Reviewer #1: Point 2 *Large-scale cascades involve failures that occur on time scales much larger than the simulation time (50 sec) considered in this manuscript, as is described in, e.g., Ref. [4] on the 2003 US blackout. It may be that many of such cascades involve a portion in which fast dynamics induces multiple line failures, but if the goal of the study is to address just that portion, it needs to be clearly stated upfront and the interpretation/conclusions should be adjusted to reflect the scope of the study. In particular, the ultimate size of the damage can only be determined by including the power plant shutdown and load shedding (which occurs on much longer time scales), so what is analyzed throughout the manuscript is an unknown fraction of the damage caused by cascades.*

Our response: We thank the reviewer for raising up this point. We agree with the reviewer that most cascading failures we cited, e.g. [1, 2, 6], took place over a total period of minutes, hours or even days, times much longer than the time period of 50 sec we simulated. However, in every report of cascading failures we are aware of, the largest damage was caused by a fast shutdown of multiple lines due to overloads. Before and after these fast events, additional power plant shutdowns and load shedding occur, contributing to the total effects of the cascade. We have now revised the “Introduction” and the “Discussion” sections of the manuscript to clarify this point and to mention explicitly which part of the cascade we are modeling. In particular, we added the sentence: “Hence, we focus throughout this manuscript on the short time scale of seconds, which contributes significantly to the overall outages” in the Introduction, and the paragraph: “Most cascades observed in the simulation are very fast, terminating on a time scale of less than 10 seconds, which supports the choice of the swing equation. Furthermore, such short time scales are consistent with empirical observations of real cascades in power grids, which were caused in a very short time by overloaded lines. Conversely, power plants (nodes of the network) were usually shut down after the failure of a large fraction of the transmission grid. The same holds for load shedding, i.e., disconnecting consumers. While the overall blackout takes place over minutes, critical damage is done within seconds due to line failures. Hence, this article models the short-time scale of line failures only.” These sentences are highlighted in blue.

In conclusion, we hope that our work, while not explaining all aspects of cascading failures, can substantially contribute to understand important factors of the intrinsic dynamics of cascades.

Reviewer #1: Point 3 *The choice of parameters, in particular, P_i and K_{ij} , seems too far from reality. Does the homogeneous choice represent any realistic situation for the Spanish grid, for example? Having a sizable generation or load on every node does not sound realistic, given that there are substations that just redistribute power coming in from one transmission line to others, and thus do not produce or consume power (if all such substations have been eliminated to generate the model, it should be stated, explained, and justified in the manuscript). These, together with all the other simplifying assumptions leading to Eq. (1), make the model used in this study too unrealistic to support the conclusions drawn and implications stated for real power grids.*

Our response: The reviewer criticizes the choice of our parameters for the simulations and questions their connection to real grids. Unfortunately, the real parameters for the Spanish or any other European grid were not available to us. Hence, we used the real topology, but an artificial choice of power (P_i) and coupling (K_{ij}) values. The simplest possible choice, adopted in the original version of the paper, was that of a homogeneous distribution. To complement the easy-to-analyze but unrealistic homogeneous coupling, we also constructed cases of heterogeneous coupling for all investigated topologies, including the Spanish grid (see Supplementary Note 1 and right column of Fig. 4 in the main text). Overall, we present investigations of homogeneous and heterogeneous coupling for the Spanish, French and British topologies in the main text and the Supplementary Information. As long as the grid is highly loaded, the cascading dynamics are very similar for all parameter settings. Furthermore, our critical link predictor works for any choice of power P_i and coupling K_{ij} .

Finally, we realize that our description of the model and its connection to the network topology was too brief in the initial submission. To elaborate: We model the power grid on a coarse-grained scale where every node may be seen as a synchronous machine with effective inertia, i.e., buses that do not generate or consume power are not present in the grids we simulate and any substations are aggregated into other nodes. This is now clearly stated in the Method section of the main manuscript.

Reviewer #1: Point 4 *I do not believe it is accurate to say that cascades studied here propagate "at a nearly constant propagation speed" with respect to the effective distance. The results in the manuscript show that the effective distance between the given line and the initial triggering line is correlated with the propagation time. However, this is not a sufficiently strong support for the description of the cascade dynamics as nearly constant propagation along the network. I believe a cascade often jumps a relatively long distance in a single step due to non-local rerouting of power flows after a line removal, and this substantially deviates from the image of a cascade wave front propagating at a constant speed (even with significant fluctuations around the mean speed).*

Our response: We thank the reviewer to point out that cascades often do not propagate with constant speed, but involve non-local rerouting of flow and thereby non-local overloads. We agree with this observation, and we have now removed the sentence “This means that the cascades propagates at a nearly constant propagation speed” from the manuscript (both from the text and from the caption of Fig.8). We have also modified the text in the revised version of the manuscript to clarify that cascades often spread non-locally in geography and that non-local effects may also be caused by voltage overloads. The new sentence, highlighted in blue, reads: “Note however that an approximately constant speed in the effective distance may correspond to highly non-local spreading in terms of geographical distances, also observed in Hines et al., and that we investigate line failures caused by current overloads in contrast to line failures due to voltage effects.”

REVIEWER #2:

Reviewer #2 *I think the paper provides a significant advance in modeling and understanding cascading failures in the grid by employing models based on "actual" grid dynamics. Recent literature in complex systems/statistical physics/networks is saturated with works claiming to model the grid and proposing to mitigate cascading failures, building on "models" which have absolutely nothing to do with the underlying dynamics of the grid (that is, conserved flows: AC model or DC approximation at its basic level). The present paper considers the AC model (swing equations), which, first and foremost, governs the actual dynamics of flows and phases on the grid. Further, this system of equations also shares some similarities with the Kuramoto model for synchronization, and thus the findings contribute to the broader understanding of synchronization dynamics. Among the main findings is that static models, in general, can significantly underestimate the size of the cascades, thus understanding the full dynamic behavior of the grid (transients on short-time scales) is important to predict and mitigate cascading failures. Also, the authors propose a flow-based predictor to identify critical lines, and an effective distance to track the propagation of the cascading failures, which seem to be working reasonably well to characterize lines and the propagation of cascades. [..., suggestions see below]*

In summary, I recommend the paper for publication with the above small changes/comments.

Our response: We thank the reviewer for the careful review of our paper. We are glad that the reviewer found our work: “a significant advance in understanding cascading failures in the grid by employing models based on actual grid dynamics”, and recommended the paper for publication. All suggested changes/comments are addressed below.

Reviewer #2: Point 1 *One comment I have is that I'm aware of some recent (independent) work of a group which have been working similar topics, namely the comparison of cascading failures in AC and DC power flow models in the grid: Hale Cetinay, Saleh Soltan, Fernando A. Kuipers, Gil Zussman, and Piet Van Mieghem, "Comparing the Effects of Failures in Power Grids under the AC and DC Power Flow Models", (Under Review at IEEE Transactions on Network Science and Engineering) <http://wimnet.ee.columbia.edu/wp-content/uploads/2017/05/comparing-the-effects-of-failures-in-power-grids-under-the-AC-and-DC-power-flow-models.pdf>. Perhaps would be best if the authors reach out to the authors above for publication information of the paper above, and mutually cite each others' work before final publication.*

Our response: We thank the reviewer for pointing us to the work of Cetinay et al. [5], which deals with similar topics, namely the differences between AC and DC models when modeling cascades. In the revised version of our manuscript, we have now included a reference to Cetinay et al. and we have discussed similarities (AC vs DC comparison) and differences. Among others things, Cetinay et al. use IEEE test grids compared to our national grid topologies. However, Cetinay et al. mainly focus on numerical treatment while we complement the numerical simulations by introducing a semi-analytic predictor for critical lines and observe the propagation of cascades.

Reviewer #2: Point 2 *My additional minor comments are: page 7: when the authors discuss the various scenarios 1., 2., 3. (for distributed or centralized power distribution with homogeneous or heterogeneous couplings), and the discussion that follows in the right column, they should indicate what Figure the panels (a),(b),(c),(d) refer to. (Figure 4, I assume.)*

Our response: We thank the reviewer for spotting this. We have now clarified this point in the revised version of the manuscript and we hope the description of the results of our statistical analysis of cascades is now clear.

Reviewer #2: Point 3 *Title: I think the model and the problem the authors study here is quite specific to the power grid (AC model/swing equations). The title with the term "supply networks" sounds a bit overly general. Perhaps replacing it with "power grids" would be more in synch with the actual system studied.*

Our response: We agree with the reviewer (see also reviewer #3, point 1) that “supply network” may sound too general. We have therefore replaced the term “supply networks” by “power grids” in the title and in several other places in the text.

REVIEWER #3:

Reviewer #3 *The authors present an extensive analysis of cascading failures in power grids via the simplified swing-equation model, thus taking into account also the dynamics. The networks analyzed are high voltage power networks, i.e. transmission networks. The paper's approach is in the spirit of statistical mechanics, hence does not pretend to give a rigorous, final answer to the problem of black-outs in real electric system but tries to understand what can trigger a systemic failure. For the network configurations and parameters analyzed, the paper shows that simulations based on event-triggered sequences of steady states tend to predict a lower number of failures respect to the more extended dynamic model of the paper. Overall, the paper is very interesting and worth of publications. I have just few of remarks for the authors*

Our response: We thank the reviewer for the careful review of our paper. We are glad that the he/she found our paper very interesting and recommended its publication. All his/her remarks are addressed below.

Reviewer #3: Point 1 *The term "supply networks" is misleading and recalls too much "supply chains" that themselves have been modeled via complex networks. Speaking of electric power grid, the correct terms are TRANSMISSION (high voltage, national & supra-national networks) and DISTRIBUTION (medium voltage, regional or big cities). The networks analysed in the paper are thus TRANSMISSION NETWORKS.*

Our response: We agree with the reviewer (see also reviewer #2, point 3) that “supply network” may be misleading. We have therefore replaced the term “supply networks” by “power grids” in the title and in several other places in the text. In the Abstract, Introduction and Discussion we have also emphasized that the focus of our work is on “transmission networks”.

Reviewer #3: Point 2 *The term transient can be also misleading if the paper is to be read by a more technical audience (power engineers and system operators). In fact, they would tend to associate a transient to something that happens at a breaker in 20-30 cycles (half a second), i.e. a factor 10 respect to the analysis of the paper; even worse, the dynamics of a transient in 20 cycles is sharp and contains higher frequencies, hence by eye inspection the time factor would be > 100 .*

Our response: The reviewer points out that “transients” can be attributed to different time scales in different communities. To avoid any possible confusion, in the revised version of the manuscript we now explicitly state the time scale (order of seconds) of our “transients”. We do so in particular in the Abstract, the Introduction and the Discussion.

Reviewer #3: Point 3 *A breaker does not trip a line instantaneously, it normally can withstand currents of order 1% of the rated currents for periods up to 1-3s. Hence, tripping a line when power hits the capacity is ok unless the derivative is flat. This point should not change the results, but perhaps could be cited.*

Our response: The reviewer points out that lines do not trip immediately when an overload occurs but may withstand it for a short time period. We thank the reviewer for this comment We have performed additional numerical simulations that take this into account. The results are reported and discussed in the new Supplementary Note 6. As predicted by the reviewer, we do not observe major changes on the statistics of cascades. In addition, we mention the finite overload time explicitly in the main text.

Reviewer #3: Point 4 *Paper [50] by Pahwa et al accounts for a cascading model amenable also of a mean field approximation; such mean field model has also been extended to interacting cascading systems in: Cascades in interdependent flow networks A Scala, PGDS Lucentini, G Caldarelli, G D'Agostino Physica D: Nonlinear Phenomena 323, 35-39 that should be added to the references in "vulnerability of multiplex networks [18-20]" Moreover, ref. "non-local effects have been considered and analyzed [8-13]" should be changed in "non-local effects have been considered and analyzed [8-13,50]" and "event-triggered sequences of steady states [8, 10, 11, 21-24]" should be changed in "event-triggered sequences of steady states [8, 10, 11, 21-24,50]"*

Our response: We thank the reviewer for pointing out additional references, which have now been included in our bibliography.

-
- [1] Bundesnetzagentur für Elektrizität, Telekommunikation und Gas. Post und eisenbahnen. bericht über die systemstörung im deutschen und europäischen verbundsystem am 4. Technical report, Technical report, German Federal Regulatory Agency for Electricity, Gas, Telecommunications, Postal and Railway Systems, Berlin, Germany, 2006.
 - [2] New York Independent System Operator. Interim report on the august 14, 2003, blackout, 2004.
 - [3] Sabine Auer, Kirsten Kleis, Paul Schultz, Jürgen Kurths, and Frank Hellmann. The impact of model detail on power grid resilience measures. *The European Physical Journal Special Topics*, 225(3):609–625, 2016.
 - [4] J. Machowski, J. Bialek, and J. Bumby. *Power system dynamics, stability and control*. John Wiley & Sons, New York, 2008.
 - [5] Cetinay, Hale and Soltan, Saleh and Kuipers, Fernando A and Zussman, Gil and Van Mieghem, Piet. Comparing the Effects of Failures in Power Grids under the AC and DC Power Flow Models. *IEEE Transactions on Network Science and Engineering*, 2017.
 - [6] Central Electricity Regulatory Commission (CERC). Report on the grid disturbances on 30th july and 31st july 2012.

Reviewers' comments:

Reviewer #1 (Remarks to the Author):

The authors did make adjustments and improvements in response to the concerns I expressed in my previous report. However, I cannot say that they have been satisfactorily addressed for concerns 2–4:

Concern 2: The authors now state that the study focuses on the impact of short-term, transient dynamics on overall outages. However, since real cascading failures occur over longer time scales (with which the authors agree), a proper study of this impact requires accurate modeling of the long-term cascading dynamics. The fact that higher-order models may predict significantly different long-term behavior (as asserted in the authors' responses) implies that some outages occurring later in the process may be missed by the model used in this manuscript. In addition, the authors attempt to substantiate their claim that the relevant outages occur within a short time period in real cascading failures by citing concrete numbers from 2006 Europe outage and 2003 US outage in their responses to my previous comments. Those numbers, however, do not prove that most of the outages occurred within a period on the order of seconds (e.g., 14 out of how many outages occurred in 14 seconds in the the Europe event?).

Concern 3: The authors now claim that the model they use is a coarse-grained model where a node may represent a group of multiple buses, collectively modeled by a single rotating machine with effective inertia. I do not see a justification of this aggregation/approximation for the Spanish grid. Does each node actually represent a group of one or more buses whose total generation is positive and can be properly modeled by a single effective generator?

Concern 4: While the non-locality of cascade propagation is now acknowledged, the authors still claim that the correlation between the effective distance and the failure arrival time allows us to define the "propagation speed" of a cascade. I believe such a notion can be meaningfully defined only in a statistical sense over an ensemble of cascades. For example, if it can be shown that the effective distance averaged over all possible triggers is monotonically increasing function of time (as is the case for, e.g., diffusion processes), then it would make sense to define the propagation speed as the derivative of that function. The current manuscript does not provide solid evidence for this. Even if that were the case, wouldn't the same hold if one uses the topological distance? For a single cascade, the effective distance as a function of the arrival time is only defined at discrete time points and jumps up and down substantially, so "speed" or derivative cannot be defined. The average speed over the whole cascade can be defined, but that would not depend on how the distance

jumps up and down, and thus can be also defined using the topological distance with no issues at all. Given all these, the claimed novelty of using the effective distance is unsubstantiated.

Reviewer #2 (Remarks to the Author):

I'm happy with the revised version, as is, for publication.

REPLIES TO THE COMMENTS OF THE REVIEWERS

We are glad to hear that the reviewers have appreciated our work. Reviewer #3 supported the paper for publication already at the first round and, with the revised version, we have satisfied the requests of Reviewer #2 (who is now also suggesting publication) and we had addressed some of the concerns of Reviewer #1.

Here is a brief summary of our comments regarding the remaining concerns of Reviewer #1, followed by more detailed.

POINT 2: The aim of our paper is to use a model that correctly captures the transient dynamics on the time scales of seconds and also allows some analytical approaches. Such a model by construction can miss additional outages that might be occurring later. However, short time scales are a key determining factor in the majority of failures, as we have now better highlighted and specified in the introduction of the manuscript. To the best of our knowledge there is no model to date providing both analytical insights and bridging the cascade events to continuous time dynamics on both short time scales and long time scales. As stated in the manuscript and in our previous replies, the work nevertheless constitutes a substantial advance because so far only very few aspects of continuous time nonlinear dynamical transients have been considered at all in studies on cascades and because short time scales are highly important to understand whether and how widely individual failures cause cascades.

POINT 3: The aggregation used in the coarse-grained model is necessary due to limited availability of information [on real-world grid data]. We describe our modelling assumptions and parameter choices in detail in the SI. If the reviewers and editors deem it important, we would be happy to move specified material from the SI to the main text.

POINT 4: We have removed every mention of "speed" as requested by the reviewer, and we point now to future in-depth work necessary to establish such a notion.

DETAILED REPLIES to the comments of Reviewer#1

POINT 2: The reviewer points out that our model, which focuses on short-time scales, may miss outages occurring later in the process and may thus miss part of the full cascading process.

We agree with the reviewer that our model focuses on short time scales and thereby does not explicitly cover all time scales and all effects that exist in cascading failures in power grids. However, we think that it is unrealistic, if not impossible, to construct a single model that covers all time scales of cascading failures at once and, at the same time, allows analytical or mechanistic insights. Indeed, even recent key work does not cover dynamics at all, see for instance the 2017 publication in Science on cascading failures in power grids, solely focusing on quasi-static analysis of cascades <http://science.sciencemag.org/content/358/6365/eaan3184>.

Our study constitutes one of the first to systematically include continuous-time dynamical transients in conjunction with and in between discrete events at all. It provides an overview on novel dynamics effects and, clear ideas about mechanisms underlying systemic phenomena occurring in cascades on the short time scales. Even if our model may miss outages occurring later in the process, it describes the majority of failures correctly on short time scales. Because most initial cascading events, as described in the introduction of the manuscript, are relevant on the time scales of seconds and thus strongly determine the evolution of the cascade, and because conceptual understanding of this subject is in its infancy, we believe that the work presented in the manuscript constitutes a valuable contribution to the literature.

If the reviewers and the editors think it can be useful to the reader, we are glad to add the following paragraph to the introduction:

"Our work complements the existing studies on cascading failures in power grids by linking nonlinear transient dynamics on short time scales to cascade events and simultaneously capturing line failures due to static overload. It is yet unrealistic to capture all aspects and time scales within a single model that is analytically tractable and provides mechanistic insights. Most of the previous studies, based on the analysis of sequences of steady states, consider the effects of power plant shutdown or line outages and did not take into account any transient dynamical effects at all. In contrast, the dynamical model we analyze provides insights into cascading failures potentially induced on short time scales, thereby characterizing the time scales relevant to the majority of line failures."

Furthermore, we apologize for the imprecision in the previous revision, where we stated that 14 links failed within 14 seconds, without mentioning the total number of outages occurred in the European blackout 2006. Indeed, a total of 33 lines failed during the European blackout and all of these events happened in a time window of 1 minute and 20 seconds. Of these 33 events, 30 lines failed in the first 19 second, and only 3 lines tripped more than 19 seconds after the initial failure. (source: https://www.entsoe.eu/fileadmin/user_upload/_library/publications/ce/otherreports/Final-Report-20070130.pdf, Page 71) In the revised manuscript, we now state this explicitly in the introduction by adding the sentence

"For example, during the European blackout in 2006, a total of 33 high voltage transmission lines tripped within a time period of 1 minute and 20 seconds, with 30 of those lines failing within the first 19 seconds."

POINT 3: The reviewer questions the aggregation/approximation of nodes into one synchronous machine for the Spanish grid.

In the previous revision we stated that our model is a coarse-grained model aggregating loads and synchronous machines in a given region as one node. To illustrate how our dynamical approach of modeling cascading failures works for real grid topologies, we have decided to use publically available information on the topology of real high-voltage transmission networks, e.g. of the Spanish grid. Notice that, while the topology of the grid is publically available, the generation and consumption at the individual nodes are not.

Hence, a reasonable possibility was to explore different types of potential power distributions, e.g.: a) many small power plants or b) fewer but larger power plants, as explained in Supplementary Note 1 and also specified in the Data Availability Statement.

To clarify this point further to the reader, we propose to include the following two paragraphs:

[When introducing the Spanish grid topology for the first time]: "We remark that, while we have complete knowledge of the network topology, due to the only partial information available on line parameters and power distribution, we have to estimate those missing parameters. Therefore, we have investigated several different power distributions and coupling scenarios in our simulations, including homogeneous versus heterogeneous coupling, as well as considering cases with many small power plants, compared to cases of fewer but larger plants. All parameter choices we have adopted are further specified in the SI and the Data Availability Statement."

Further to address the question by the reviewer "Does each node actually represent a group of one or more buses whose total generation is positive and can be properly modeled by a single effective generator?" [The existing sentence on synchronous machines in the Method section reads.] 'Note that this is a coarse-grained model where every node is modeled as a rotating machine with effective inertia.' [We would like to add the following sentence to address the question.] "A node with higher demand than supply will then act as an effective consumer, i.e., a synchronous motor."

POINT 4: The reviewer criticizes our claim that the correlation between the effective distance and the failure arrival time allows to define the "propagation speed" of a cascade. The reviewer finds that such a notion of velocity can only be meaningfully defined in a "statistical sense" by averaging over all possible triggers.

If we understand the reviewer correctly, a statistical evaluation is indeed what we have already done in the original version of our work. The results were reported in Supplementary Figure 7 of the SI. We considered all possible triggers and, for each one of them, we computed the correlation between distance and time of arrival of the failure. Instead of averaging correlation and slope (presumed velocity) over all the events, we have shown the entire distribution of regressions coefficients and slopes/velocities in Supplementary Figure 7, panels c and d.

The figure shows that the correlation between distance and arrival time is higher when effective distances is used in contrast to topological distances.

We still agree with the reviewer that further investigations would be needed to meaningfully and robustly define a notion of velocity of the cascade. So we have decided to remove any mentions of "propagation speed" from the main manuscript. The definition of a propagation speed will only be briefly discussed in the discussion section as a future possible project. With our work we hope to stimulate the research community to include dynamical effects in cascades and consider alternative distance measure in future investigations.

We highlight the removal of "speed" in the text by strikethrough commands, replacing previous statements by referring to correlation. To highlight future investigations, we added the following sentences in the discussion:

"The observed correlation between propagation time and effective distance of a failure points to the possibility of extracting an effective speed of the cascade propagation; this result may thus stimulate further research understanding propagation patterns on networks. Being able to measure the speed of a cascade would further contribute to the design of measures to stop or contain cascades in real time because such propagation speed determines how fast actions have to be taken."

REVIEWERS' COMMENTS:

Reviewer #1 (Remarks to the Author):

The two rounds of reviews and revisions have clarified the limitations of the study, and I feel that the level of significance and the breadth of impact of the results are not sufficient for a publication in a top journal like Nature Communications. For such a publication, it is important that the results are either based on realistic models or validated against real systems. The study aims "to use a model that correctly captures the transient dynamics on the time scales of seconds and also allows some analytical approaches." I now clearly see that the model captures only the initial part of each cascade for which the (necessary) short-term assumption is valid. Additional, more thorough analysis to quantify the extent to which it captures the failures is needed. I am also concerned that some additional failures are missed because of the transmission lines that are eliminated in the process of the aggregation / coarse-graining. Overall, while the model provides a starting point for understanding the impact of transient dynamics on cascades, I feel that it lacks the level of realism needed to make the results sufficiently meaningful/interesting to the broad audience of Nature Communications.

Also, I have a similar opinion on the idea of "propagation speed" of a cascade. If plotting with respect to a notion of distance were to give a more deterministic curve (with much less vertical variations than in Fig. 8 and Supp. Fig. 7), then it would give an interesting insight. But this is not the case with the effective distance considered in the manuscript. There, the result is that there is somewhat higher correlation compared to using the graph distance, which gives some insight but much less than with the deterministic description. I do not believe this is sufficient to justify publication in a journal at this level.

Reviewer #3 (Remarks to the Author):

POINT 1: The introduction will benefit from the proposed paragraph "Our work complements the existing studies"

Regarding POINT 3, I think that all the clarifications proposed are fair and should be implemented

Regarding point 4: the quasistatic analysis of the paper

<http://science.sciencemag.org/content/358/6365/eaan3184> shows that individual cascades often propagate far from the triggering failures. In general, the long range character of cascades is present not only in static models of power flow cascades, but also in the records of severe black outs like the 10 August 1996 WCCA blackout (see page 42-43 of the report "1996 System Disturbances - Review of Selected 1996 Electric System Disturbances in North America") As an example, while the first 17 events happen in a close geographical area, event 18 happens in a far away region of the map. Also notice that the time separation among events is of the order of few minutes. Authors should discuss the possibility that their model complements and integrates static ones.

REPLIES TO THE REMAINING COMMENTS OF THE REVIEWERS

We are happy that two of the three reviewers recommend publication in *Nature Communications* and that the Editors are happy to publish a suitably revised version of our manuscript.

Below, we address the remaining comments raised by the reviewers on a point-by-point basis.

Reviewer #1

The two rounds of reviews and revisions have clarified the limitations of the study, and I feel that the level of significance and the breadth of impact of the results are not sufficient for a publication in a top journal like Nature Communications. For such a publication, it is important that the results are either based on realistic models or validated against real systems. The study aims "to use a model that correctly captures the transient dynamics on the time scales of seconds and also allows some analytical approaches." I now clearly see that the model captures only the initial part of each cascade for which the (necessary) short-term assumption is valid. Additional, more thorough analysis to quantify the extent to which it captures the failures is needed. I am also concerned that some additional failures are missed because of the transmission lines that are eliminated in the process of the aggregation / coarse-graining. Overall, while the model provides a starting point for understanding the impact of transient dynamics on cascades, I feel that it lacks the level of realism needed to make the results sufficiently meaningful/interesting to the broad audience of Nature Communications.

Our response: We agree with the reviewer that our study does have limitations and does not cover all aspects of cascading failures. We are confident that the changes made in the previous revision clarify the scope of our work to the readership. The revised manuscript includes specific statements about time scales and non-locally covered and discusses their relevance in real world outages.

Also, I have a similar opinion on the idea of "propagation speed" of a cascade. If plotting with respect to a notion of distance were to give a more deterministic curve (with much less vertical variations than in Fig. 8 and Supp. Fig. 7), then it would give an interesting insight. But this is not the case with the effective distance considered in the manuscript. There, the result is that there is somewhat higher correlation compared to using the graph distance, which gives some insight but much less than with the deterministic description. I do not believe this is sufficient to justify publication in a journal at this level.

Our response: As stated in the previous revision, we removed the notion of a "propagation speed" and instead point to future research to investigate the increased correlation when using effective distance. A purely deterministic description, as the one the reviewer asks for, may not exist. The issue is subject of ongoing and future research.

Reviewer #3

POINT 1: The introduction will benefit from the proposed paragraph "Our work complements the existing studies"

Our response: We included the suggested paragraph in the final version.

Regarding POINT 3, I think that all the clarifications proposed are fair and should be implemented

Our response: We implemented the changes as suggested in the final version.

Regarding point 4: the quasistatic analysis of the paper <http://science.sciencemag.org/content/358/6365/eaan3184> shows that individual cascades often propagate far from the triggering failures. In general, the long range character of cascades is present not only in static models of power flow cascades, but also in the records of severe black outs like the 10 August 1996 WCCA blackout(see page 42-43 of the report "1996 System Disturbances - Review of Selected 1996 Electric System Disturbances in North America") As an example, while the first 17 events happen in a close geographical area, event 18 happens in a far away region of the map. Also notice that the time separation among events is of the order of few minutes. Authors should discuss the possibility that their model complements and integrates static ones.

Our response: Our study indeed points to the potential of locally propagating cascades caused by transient current overloads and thus complements previous studies carried out using quasi-static models. We discuss how we complement these approaches in the final version of the manuscript by including the following statement

"Moreover, propagation patterns of line failures caused by current overloads, as investigated here, may be qualitatively different from those caused by voltage effects [4, 58]. On longer time scales the operation of control systems and emergency measures such as load shedding must be taken into account to assess the impact of a cascade of failures. These features are typically studied in quasi-static models such that the short time scale considered in this paper offers a complementary view to the spreading of cascading failures."

The discussion explicitly mentions voltage effects and emergency measures which are not included in our work and may lead to strongly nonlocal effects. We remark that the first nonlocal effect during the August 96 blackout happened only after the fragmentation of the grid into four islands. The frequency in the entire Northern island increased to over 90.4 Hz such that every node in the island was affected at this stage.